# Probing TDP-43 condensation using an in silico designed aptamer

Elsa Zacco 1,12, Owen Kantelberg2,12, Edoardo Milanetti3,4, Alexandros Armaos1, Francesco Paolo Panei3, Jenna Gregory 5,6,7, Kiani Jeacock2, David J. Clarke 2, Siddharthan Chandran5,6,7, Giancarlo Ruocco3,4, Stefano Gustincich 1, Mathew H. Horrocks 2✉, Annalisa Pastore 8✉ & Gian Gaetano Tartaglia 1,9,10,11✉

Aptamers are artificial oligonucleotides binding to specific molecular targets. They have a promising role in therapeutics and diagnostics but are often difficult to design. Here, we exploited the *cat*RAPID algorithm to generate aptamers targeting TAR DNA-binding protein 43 (TDP-43), whose aggregation is associated with Amyotrophic Lateral Sclerosis. On the pathway to forming insoluble inclusions, TDP-43 adopts a heterogeneous population of assemblies, many smaller than the diffraction-limit of light. We demonstrated that our aptamers bind TDP-43 and used the tightest interactor, Apt-1, as a probe to visualize TDP-43 condensates with super-resolution microscopy. At a resolution of 10 nanometers, we tracked TDP-43 oligomers undetectable by standard approaches. In cells, Apt-1 interacts with both diffuse and condensed forms of TDP-43, indicating that Apt-1 can be exploited to follow TDP-43 phase transition. The de novo generation of aptamers and their use for microscopy opens a new page to study protein condensation.

[1] Centre for Human Technologies (CHT), Istituto Italiano di Tecnologia (IIT), Via Enrico Melen, 83, 16152 Genova, Italy. [2] EaStCHEM School of Chemistry, University of Edinburgh, Edinburgh EH9 3FJ, UK. [3] Department of Physics, Sapienza University, Piazzale Aldo Moro 5, 00185 Rome, Italy. [4] Center for Life Nanoscience, Istituto Italiano di Tecnologia, Viale Regina Elena 291, 00161 Rome, Italy. [5] UK Dementia Research Institute at University of Edinburgh, University of Edinburgh, Edinburgh bioQuarter, Chancellor's Building, 49 Little F, Edinburgh, UK. [6] Centre for Clinical Brain Sciences, University of Edinburgh, Edinburgh, UK. [7] Euan MacDonald Centre for MND Research, University of Edinburgh, Edinburgh, UK. [8] UK Dementia Research Institute at the Maurice Wohl Institute of King's College London, London SE5 9RT, UK. [9] Centre for Genomic Regulation (CRG), Dr. Aiguader 88, 08003 Barcelona, Spain. [10] Catalan Institution for Research and Advanced Studies, ICREA, Passeig Lluís Companys 23, 08010 Barcelona, Spain. [11] Department of Biology 'Charles Darwin', Sapienza University of Rome, P.le A. Moro 5, Rome 00185, Italy. [12] These authors contributed equally: Elsa Zacco, Owen Kantelberg. ✉email: mathew. horrocks@ed.ac.uk; annalisa.pastore@crick.ac.uk; gian.tartaglia@iit.it

Aptamers are short DNA or RNA oligonucleotides that bind specific targets with high affinity[1]. Similar to antibodies, aptamers represent a powerful tool for medical applications, especially diagnostics[2]. Aptamers, however, have a number of advantages over their protein counterparts[3]: once an aptamer sequence is identified, they are easier and less expensive to prepare and chemically modify. They have the potential to bind to a wider range of targets, from ions and small molecules to proteins and large complexes.

Despite their undiscussed value, one of the main drawbacks of aptamers is their generation. To date, experimental approaches such as the systematic evolution of ligands by exponential enrichment (SELEX) technique with its variants and improvements[4] stand at the basis of the development of therapeutic aptamers[5] through high throughput screening of oligonucleotide libraries. This approach, however, has several limitations: it needs careful planning, specific expertise, and is costly and time consuming. Successful selection heavily depends on the quality and coverage of the nucleic acid library employed[2,6]. Last, SELEX cannot ensure specificity, as it selects RNA sequences for one target without considering interactions that might occur with other molecules.

Here, we demonstrate that we can rationally design aptamers able to bind a specific protein using an in silico approach. We have previously developed a method that can perform large-scale predictions of protein–RNA interactions. The algorithm, named *cat*RAPID, predicts interactions between protein and RNA pairs based on the physico-chemical properties encoded in their sequences[7,8]. As previously shown in our works, the interaction propensity calculated with *cat*RAPID correlates with the experimental binding affinities[9,10] and was successfully exploited to identify the binding partners of non-coding transcripts such as *Xist*[8], *HOTAIR*[11,12], *SAMMSON*[13], in addition to interactomes of RNA genomes[14].

In the present work, we describe how we stretched the *cat*RAPID algorithm to design short RNA aptamers, by flanking it with a whole pipeline of computational tools. As a proof-of-concept, we demonstrate that we can generate high-affinity RNA aptamers targeting the TAR DNA-binding protein 43 (TDP-43)[15]. This is a modular RNA-binding protein, whose architecture comprises an N-terminal domain, two RNA recognition motif (RRM) domains that preferentially bind single-stranded GU-rich RNA sequences[16], and a low-complexity C-terminus. TDP-43 is heavily implicated in amyotrophic lateral sclerosis (ALS), frontotemporal dementia (FTD), and Alzheimer's disease (AD), relentlessly progressive neurodegenerative diseases with no cure[17]. Mislocalization of nuclear TDP-43 to the cytoplasm and its subsequent aggregation into toxic inclusions is a common pathological feature in over 97% of all ALS cases[18]. Along the pathway to forming such insoluble inclusions, TDP-43 adopts a highly heterogenous population of phase-separated assemblies with differing sizes and structures, which are often smaller than the diffraction limit of light, thus limiting their characterization by standard fluorescence microscopy.

The diffraction-limit of light restricts optical microscopy to a resolution of ~250 nm. Recently, numerous techniques, grouped under the umbrella term of super-resolution (SR) microscopy[19], have been developed to surpass this limit, enabling imaging at a resolution as high as 5 nm. We have previously developed an SR method that makes use of aptamers to image targets with nanometer resolution[20]. Aptamer DNA-PAINT (AD-PAINT), much like DNA-PAINT[21], is a pointillism-based SR technique employing temporal separation of fluorescent localizations to gain a higher spatial resolution. A short single-stranded "docking" DNA oligonucleotide is attached to a target-specific probe capable of binding to the molecule-of-interest. Visualization of the probe occurs when a second single-stranded "imaging" DNA strand tagged with an organic fluorophore transiently binds to the docking sequence and its location is determined with nanometer precision. The repeated binding and unbinding of multiple probes allow SR to be achieved.

In this study we exploit the transient binding of an in silico designed aptamer, Apt-1, to TDP-43 to mimic the transient binding of the "docking" and "imaging" strands, thus allowing the use of Apt-1 labeled with an organic fluorophore for SR imaging and eliminating the two-component system of AD-PAINT. We have previously used AD-PAINT to visualize aggregates of Aβ42 and α-synuclein[20], exploiting an aptamer that did not bind to a specific protein, but recognized instead the structural motif of the extended beta-sheet found in amyloid aggregates. With the possibility to generate TDP-43-specific aptamers through the *cat*RAPID algorithm, we thus gain the unique advantage to have aptamers that can be used to image the condensation of a selected protein without interferences from other components with similar structural characteristics.

We confidently predict that the aptamers generated for TDP-43 will provide new detailed insights into the aggregation process of TDP-43. More in general, our in silico design method provides a unique tool to assist quite different applications and may become a standard technique to select RNA aptamers offering, for instance, a new perspective for drug design.

## Results

**Design of RNA aptamers binding to TDP-43**. We first hypothesized that *cat*RAPID, a software that has been already proven to successfully identify RNA–protein interactions[7,8], could be used to design RNA aptamers against TDP-43. Starting from an iCLIP library of $3 \times 10^6$ nucleotides found associated with TDP-43[16], we evaluated to what extent physical interactions could be identified by *cat*RAPID. The algorithm discriminates interacting and non-interacting regions with performances that increase proportionally to the iCLIP score, indicating that strong-signal regions could be accurately predicted (Fig. 1a; Online Methods). Since TDP-43 preferentially binds GU-rich RNA through its RRM domains[16], we counted the number of GU repetitions in the top and bottom 100 ranked regions based on the iCLIP scores (corresponding to the strongest and poorest 1‰ signal sequences). We found a significant increase in the number of GU repeats (Fig. S1; *p*-value $4 \times 10^{-2}$, Wilcoxon signed-rank test). Stronger enrichment was found when ordering all the transcriptomic regions according to the *cat*RAPID score (Fig. S1; *p*-value $2 \times 10^{-6}$, Wilcoxon signed-rank test), which indicated an enrichment of potential TDP-43 targets.

Starting from the top-interacting transcriptomic regions (Fig. 1a; Online Methods), we generated a list of fragment sequences using a window of 10 nucleotides moved from the 5′ to the 3′ of each sequence (Table S1). The length of the window was set according to the number of contacts that an RNA oligonucleotide establishes with the RRM domains of TDP-43 in an NMR structure (PDB 4bs2; Fig. 1b; Online Methods)[22]. A size of 10 nucleotides also ensures that most of the fragments are single-stranded, which is a requirement for TDP-43 binding[23,24]. For each list of fragments, we computed the *RNA Fitness* and *Protein Fitness* scores using *cat*RAPID (Online Methods). The *RNA Fitness* score was defined as the value obtained by randomly introducing mutations in the RNA sequences to alter the interaction propensity for TDP-43 (Fig. 1c; Online Methods). The *Protein Fitness* or specificity score was built to evaluate the strength of the interaction propensity of each RNA sequence for TDP-43 as compared to a pool of polypeptide chains of the same length and amino acid composition (Fig. 1d). Both the *RNA* and

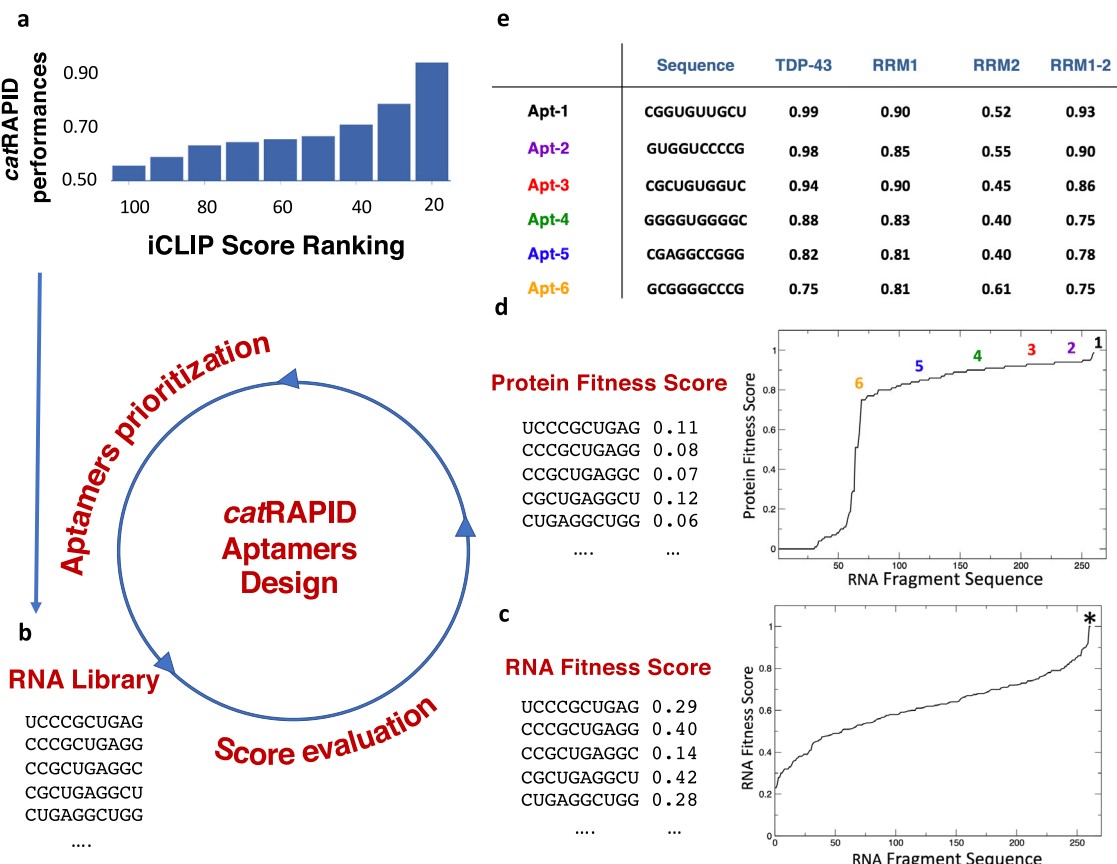

**Fig. 1 In silico design of RNA aptamers.** Sketch of the computational pipeline. **a** TDP-43 interaction propensity computed with catRAPID is used to discriminate between interacting (high iCLIP score) and non-interacting (low iCLIP scores) transcriptomic regions. The iCLIP scores were ranked from low (100 transcriptomic regions with the highest iCLIP score vs 100 transcriptomic regions with the lowest iCLIP score) to high (20 transcriptomic regions with the highest iCLIP score vs 20 transcriptomic regions with the lowest iCLIP score). catRAPID performances increase with the strength of the iCLIP score, indicating that the algorithm can accurately identify strong-signal interactions. **b** Using the 30 top-interacting transcriptomic regions we generated a list of fragments using a sliding window of 10 nucleotides moved from the 5′ to 3′ of each sequence. **c** The *RNA Fitness* score was employed to measure the strength of catRAPID interaction propensities upon mutation of the RNA fragment sequence. Examples of RNA fragments and their relative *RNA Fitness* scores are shown on the left panel. On the right panel, the *RNA Fitness* scores of all the RNA fragments generated are given (candidate aptamers were marked with a star). **d** The *Protein Fitness* score measures the strength of interaction propensity of each RNA sequence for TDP-43 in comparison with a pool of protein sequences with the same length and amino acid composition. Examples of RNA fragments and their *Protein Fitness* scores are shown on the left panel. On the right panel, the *RNA Fitness* scores of the RNA fragments generated are given (candidate aptamers were marked with different numbers and colors). **e** Summary of *Protein Fitness* scores for TDP-43 full-length, RRM1, RRM2, RRM1-2 sequences (the colors match the Protein Fitness reported in panel **d**).

*Protein Fitness* scores ranged between 0 (low rank) and 1 (high rank) and were calculated for the interaction between each aptamer and full-length TDP-43 sequence.

By considering the *RNA* and *Protein Fitness* scores, we prioritized a set of RNA sequences: six candidate aptamers were selected with an *RNA Fitness* score of 1 and a *Protein Fitness* score ranging from 0.75 to 0.99 (Fig. 1e). Thus, the selected sequences had *RNA Fitness* that could not be further increased by mutating their nucleic acid sequences, and a specificity for TDP-43 ranging from low (Apt-6, value of 0.75; Fig. 1e) to high (Apt-1, value of 0.99; Fig. 1e). We note that the *Protein Fitness* scores calculated on full-length TDP-43 have a Pearson correlation of 0.88 with *Protein Fitness* scores calculated for the two RRM domains (amino acids 102–269, hereafter named RRM1-2) and a correlation of 0.81 with the *Protein Fitness* scores calculated on the RRM1 alone (negligible correlation with RRM2; Fig. 1e). This is interesting because it is known that RRM1 is necessary and sufficient for the interaction, whereas RRM2 is unable to bind in isolation[23,25]. The presence of RRM2 in tandem with RRM1 cooperatively increases the affinity, thanks to additional interactions established within RRM2[24]. Thus,

our candidate aptamers were predicted to specifically interact with RRM1-2 and RRM1.

Taken together, these data suggest that our in silico method can, in principle, provide protein-specific predictions of RNA-binding and rank the results in terms of their expected binding propensity.

**Experimental validation of TDP-43 RNA-binding regions with RNA aptamers.** We validated the aptamer binding affinities by measuring the dissociation constants ($K_d$) through bio-layer interferometry (BLI). In this analysis, we used a construct containing the RRM1-2 domain where *cat*RAPID indicated specific binding of the aptamers (Fig. 1e). Importantly, RRM1-2 represents the region necessary for RNA-binding with high affinity[22]. Isolated RRM1-2 is purifiable as a monomer under near-to-physiological conditions and in suitable quantities for biophysical studies[24].

We first analyzed the binding affinities of the individual RRM1 (residues 102–191) and RRM2 (residues 191–269) domains. We

found that RRM1 strongly interacts with the aptamers, with $K_d$ values in the high nanomolar to low micromolar range, whereas the isolated RRM2 does not appreciably bind under our experimental conditions (Table 1). This result is in agreement with *cat*RAPID predictions (Fig. 1e) and with previous literature[23,24]. A significant improvement of the affinity for almost all aptamers was observed with RRM1-2. Apt-1 has the highest affinity, with a $K_d$ of 100 nM (Figs. 2a and S2). As in a previous study[26], we employed the reverse complementary RNA (nApt-1) as a negative control (with a *Protein Fitness* score of 0.18 and an *RNA Fitness* score of 0.60). The negative control has a $K_d$ of 1.5–2.0 μM (Figs. 2b and S2), which is in the range of values obtained for Apt-4, Apt-5, and Apt-6 (Table 1). The experimental $K_d$ values correlate with the predicted *Protein Fitness* scores computed by *cat*RAPID (Fig. 2c; Fig. 1e; Pearson correlation of −0.93). As a control, we investigated the interaction of Apt-1 and of nApt-1 with two more amyloidogenic proteins in their soluble forms: Aβ42 and α-synuclein (Fig. S3). Neither protein showed binding to Apt-1 or nApt-1 within the tested conditions, emphasizing the specificity of Apt-1 towards TDP-43.

These results validate the computational design of RNA aptamers and indicate Apt-1 as the sequence with the highest affinity for TDP-43. They also demonstrate that we can confidently rely on *cat*RAPID predictions.

**Dynamics of Apt-1 interactions with TDP-43**. We used molecular dynamics (MD) to investigate the structural mechanism of TDP-43 interaction with Apt-1 and its control nApt-1 (Table S2). The SimRNA software was first used to generate an ensemble of initial conformers[27], short MD trajectories were then run to generate structures for docking through HDOCK[28]

and full MD trajectories in the microsecond scale were produced (Online Methods). MD calculations were carried out at 300 K in explicit solvent. Different descriptors were used to study protein–RNA interactions at equilibrium: the number of contacts that Apt-1 and nApt-1 establish with RRM1-2, the distances and hydrogen-bond (H-bond) distributions as well as the covariance of atoms motions over trajectories (Fig. S4). The number of contacts, distance, and numbers of H-bonds provide an estimate of the tightness of contacts[29]. The average motion covariance informs on the atomic fluctuations of the binding site at the interface[30] (Online Methods; Table 2). We carried out MD simulation of RRM1-2 in complex with an RNA oligonucleotide to build a reference for our analyses (Online Methods)[22].

Our MD simulations indicate that RRM1-2 interactions with Apt-1 (number of contacts: 38, contacts distance: 6.29 Å, fraction of H-bonds at 2.9 Å or below: 50%) are comparable to those identified within the NMR structure used as reference (number of contacts: 36, contacts distance: 4.69 Å, fraction of H-bonds at 2.9 Å or below: 47%). This structure offers an ideal reference set, since it defines the interactions between the same domains of TDP-43 (RRM1-2) and a short RNA sequence natural binder of TDP-43, according to CLIP data[22]. In addition, we verified, in silico and in vitro, the structural similarity between Apt-1 and the short RNA reported in the NMR structure and confirmed Apt-1 to be linear (Fig. S5). Fewer contacts were found for nApt-1 (number of contacts: 30, contacts distance: 8.70 Å, fraction of H-bonds at 2.9 Å or below: 42%; Fig. S6; Online Methods). Similarly, Apt-1 was associated with strongly positive covariance of binding site motions (0.56) similar to those of the NMR oligonucleotide (0.76), indicating stable interactions. By contrast, nApt-1 showed a negative correlation (−0.53), indicating asynchronous motions and more unstable interactions (Table 2).

Both aptamers form contacts with amino acids 180 and 224 that are also present in the reference NMR model structure (PDB 4bs2; Fig. 3a). Yet, Apt-1 interacts more frequently with amino

---

**Table 1 Binding affinities of the RNA aptamers for RRM1 and RRM1-2.**

|  | $K_d$ screening (μM) | |
| --- | --- | --- |
|  | RRM1 | RRM1-2 |
| Apt-1 | 0.58 ± 0.01 | 0.10 ± 0.01 |
| Apt-2 | 1.44 ± 0.40 | 0.75 ± 0.15 |
| Apt-3 | 0.90 ± 0.20 | 0.65 ± 0.12 |
| Apt-4 | 0.90 ± 0.35 | 1.30 ± 0.45 |
| Apt-5 | 1.65 ± 0.30 | 1.50 ± 0.30 |
| Apt-6 | 2.50 ± 0.65 | 1.60 ± 0.40 |

$K_d$ values of candidate RNA aptamers interactions with the isolated RRM1 of TDP-43 (second column) and RRM1-2 (third column; mean ± S.D.; $n = 3$).

---

**Table 2 For Apt-1 and its negative control nApt-1 we computed the RRM1-2 contact distance (first column), distance of RRM1-2 contacts (second column), fraction of H-Bonds (<2.9 Å, third column), and motion covariance (fourth column) over MD trajectories at equilibrium.**

|  | RRM1-2 contacts | Contacts distance (Å) | H-bonds (2.9 Å) | Motion covariance |
| --- | --- | --- | --- | --- |
| Apt-1 | 38 | 6.29 | 50% | +0.56 |
| nApt-1 | 28 | 8.70 | 43% | −0.53 |

---

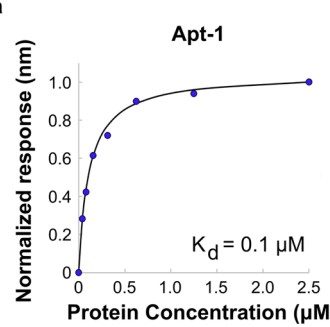

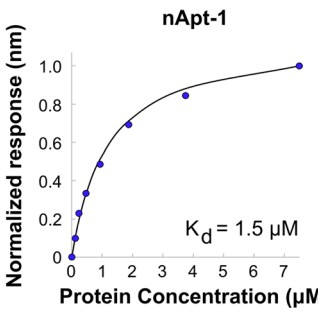

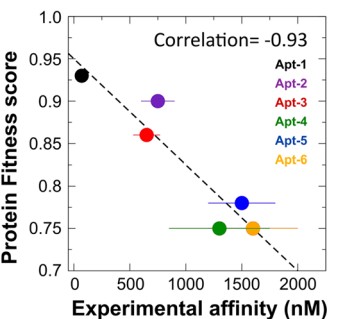

**Fig. 2 In vitro validation of aptamers binding affinity. a** Binding response curve for the interaction between RRM1-2 and Apt-1. **b** Binding response curve for the interaction between RRM1-2 and negative RNA (antisense Apt-1). **c** Experimental $K_d$ and *Protein Fitness* scores are tightly anti-correlated (Pearson correlation of −0.93). The experimental values are reported in Table 1. Source data are provided as a Source data file[64].

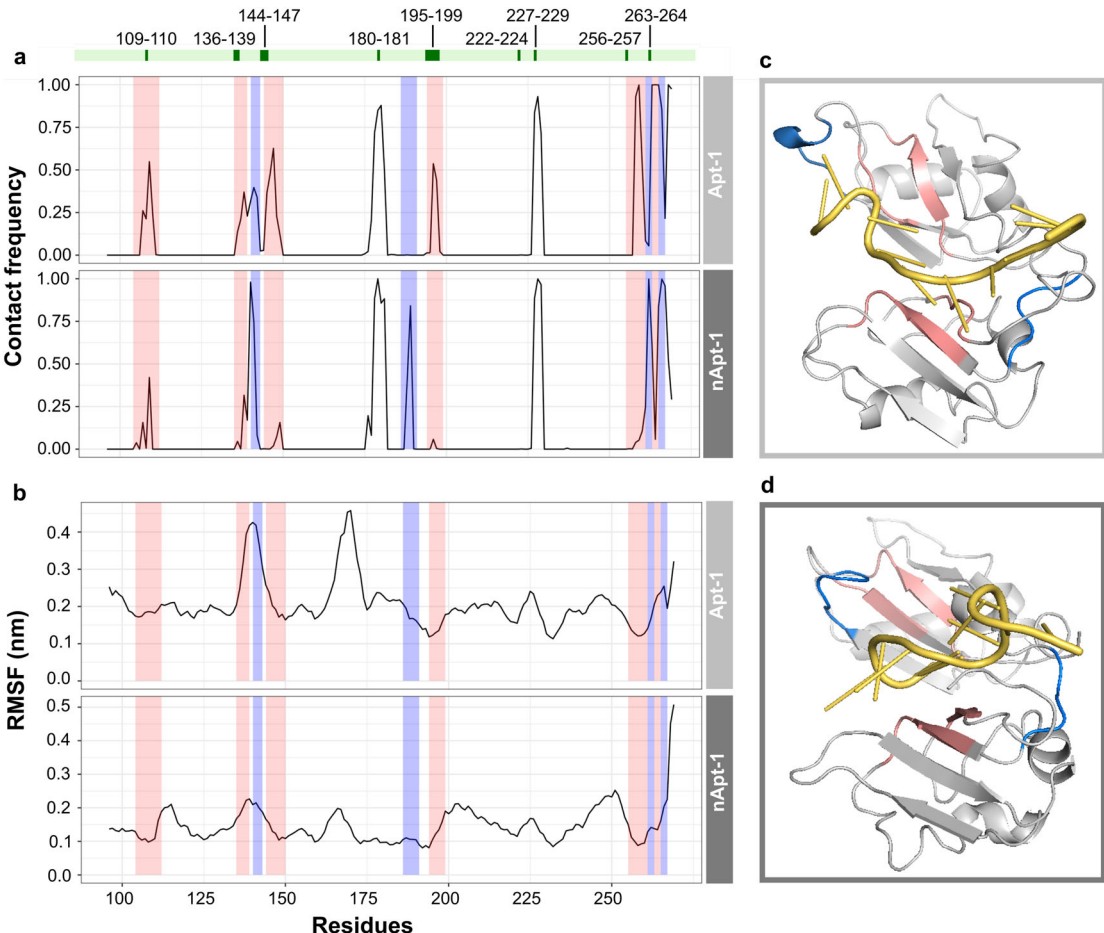

**Fig. 3 MD characterization of Apt-1 and nApt-1 interactions with RRM1-2. a** Contact frequency computed along Apt-1 and nApt-1 MD trajectories; the contacts observed in the NMR model structure are marked with green boxes at the top of the image. Red is used to indicate contacts more frequent for Apt-1 and blue contacts that are more frequent for nApt-1. **b** Root mean square fluctuation computed along Apt-1 and nApt-1 MD trajectories. The color of the boxes follows the same as in panel (**a**). **c** Structural representation of residues with stable contacts in complex with Apt-1 (RRM1-2 representative configuration at equilibrium shown; colors correspond to panels (**a**) and (**b**). **d** Structural representation of residues with stable contacts in complex with Apt-1 (RRM1-2 representative configuration at equilibrium shown; colors correspond to panels (**a**) and (**b**).

acids 104–112,135–139,144–150, 194–199, 255–261, and 263–264 (Fig. 3a), as reported in the NMR model, while nApt-1 interacts with amino acids 140–143,165,186–191, 262–263, and 267 (Fig. 3a) that are not involved in the binding in the NMR model. The Root Mean Square Fluctuation (RMSF; Fig. 3b; Online Methods) shows that nApt-1 binds less well because the contacts are formed in highly mobile regions (amino acids 144 and 267, corresponding to loops), whereas Apt-1 interacts with elements with lower flexibility (amino acids 135, 145, 255, and 263, Fig. 3b–d).

In summary, MD confirms that Apt-1 binds tightly to RRM1-2 in the same regions that were previously reported in NMR models, thus supporting the reliability of *cat*RAPID predictions.

**Using Apt-1 as a super-resolution imaging probe for TDP-43 aggregates**. Having shown that Apt-1 is able to bind to RRM1-2, we next explored the possibility of using this aptamer as imaging probe to monitor the aggregation of TDP-43. Indeed, aptamers can be used for a number of applications, including the visualization of specific targets[20] and their soluble or aggregated state[31]. We previously demonstrated that a molar excess of the NMR oligonucleotide added to soluble RRM1-2 at the beginning of a kinetics of aggregation can reduce its propensity to undergo condensation[31]. To determine whether Apt-1 could interfere with

already formed aggregates, we added the aptamer 24 h after initiation of the aggregation. We observed that Apt-1 has no detectable effects on the aggregation kinetics (Fig. S7). This proves that Apt-1 is a suitable ex-situ imaging probe for RRM1-2 aggregation without disturbing their assembly.

We exploited Apt-1 to track TDP-43 aggregation (Fig. 4a, b). Full-length TDP-43 typically forms oligomers of 10–1000 nm[32], but, due to the diffraction-limit of light, their visualization is restricted to 250 nm (Fig. 4c–e). We modified Apt-1 for SR microscopy, which can be employed to achieve a resolution of 5 nm (Fig. 4d)[33]. More specifically, we labeled Apt-1 with the organic fluorophore Alexa Fluor 590 and added it at a concentration of 1 nM to a series of samples taken from a solution of RRM1-2 incubated under conditions favouring its aggregation (Online Methods). Unlike AD-PAINT[20], which requires the aptamer to be modified with a short DNA sequence for DNA-PAINT, the affinity of Apt-1 to RRM1-2 meant that it was possible to localize and build up a SR image as the labeled aptamer transiently binds to the aggregates (Fig. 4f, g). We also made use of the amyloid-binding dye ThT and single-aggregate visualization by enhancement (SAVE) imaging to determine the extended beta-sheet content of each oligomer detected by SR microscopy (Figs. 5a and S8). At the start of the aggregation reaction, pre-aggregates of RRM1-2 were detected; however, these

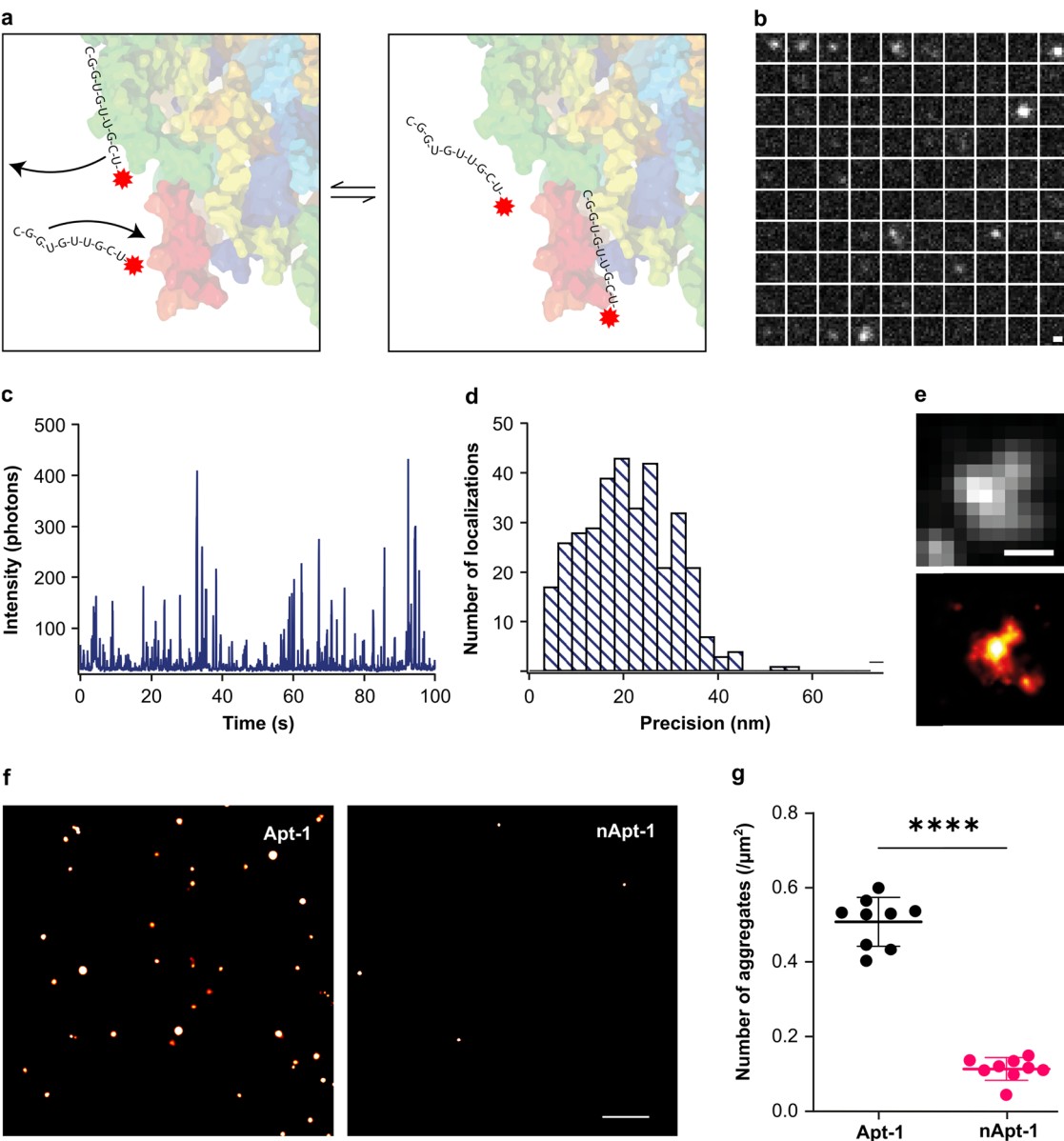

**Fig. 4 SR imaging of surface-immobilized RRM1-2 aggregates. a** Schematic representation of SR imaging. The immobilized RRM1-2 aggregate is transiently bound by an Atto590-tagged Apt1 molecule, the position of which is determined with nanometer precision. This process is repeated to generate a SR image of each aggregate. **b** Example time montage of an oligomer being imaged using PAINT. Each sub-image is separated by 1 s, moving through time from left to right then top to bottom; scale bar: 400 nm. **c** Intensity profile of the oligomer imaged in (**b**). Individual localizations appear as bursts in intensity that are separated in space and time. **d** Histogram of precisions of the oligomer imaged in (**b**). Each localization is accurate positioned with a precision of 60 nm or less. **e** SR image (red hot) and diffraction-limited (gray) images of the aggregate shown in (**b**), scale bar is 500 nm. **f** Sample fields of view of clustered RRM1-2 aggregates imaged with Apt-1 and nApt-1, scale bar 1 μm. Data from 3 independent experiments. **g** Compared to Apt-1 nApt-1 detects significantly less RRM1-2 aggregates. The data shown are mean ± SD of 9 fields of view. ****$p = 1.9769 \times 10^{-11}$; analyzed by two-sided $t$-test, confidence interval 95%. Source data are provided as a Source data file[64].

decreased in number in the first 4 h of the aggregation reaction (for example Apt-1 and SAVE images, see Fig. 5). There was subsequently a sharp increase in the number of aggregates detected using Apt-1 before eventually plateauing and decreasing at 72 h (Fig. 5b). Despite the rapid increase in the number of aggregates detectable using Apt-1, the number of ThT-active aggregates increased slowly (Fig. 5c) over 72 h, as did the number of aggregates that bound both Apt-1 and ThT (Fig. 5d) (Fig. S9 shows the fraction of Apt-1 aggregates that were ThT-active during the aggregation reaction). In a similar fashion to alpha-synuclein in Parkinson's disease[34,35], amyloid-beta in Alzheimer's disease[36], and aggregation-prone proteins in prion diseases[37],

TDP-43 might have the potential to form aggregates with different morphologies and structures, the more mature one containing extended beta-sheet structure, as shown here imaging RRM1-2. The mean number of localizations per aggregate (Fig. 5e) increased as the mean length and areas of the aggregates (Fig. 5f and g) increased over 72 h (see Figs. S10–S12 for localization number, length, and area histograms for each time-point). The resolution values for each image are reported in Table S3. To compare the sequence specificity of Apt-1 to TDP-43, sample images of RRM1-2 aggregates were taken with Apt-1 or nApt-1. The number of aggregates detected with Apt-1 was significantly higher than with nApt-1 (Fig. S13; $p$-value < 0.0001,

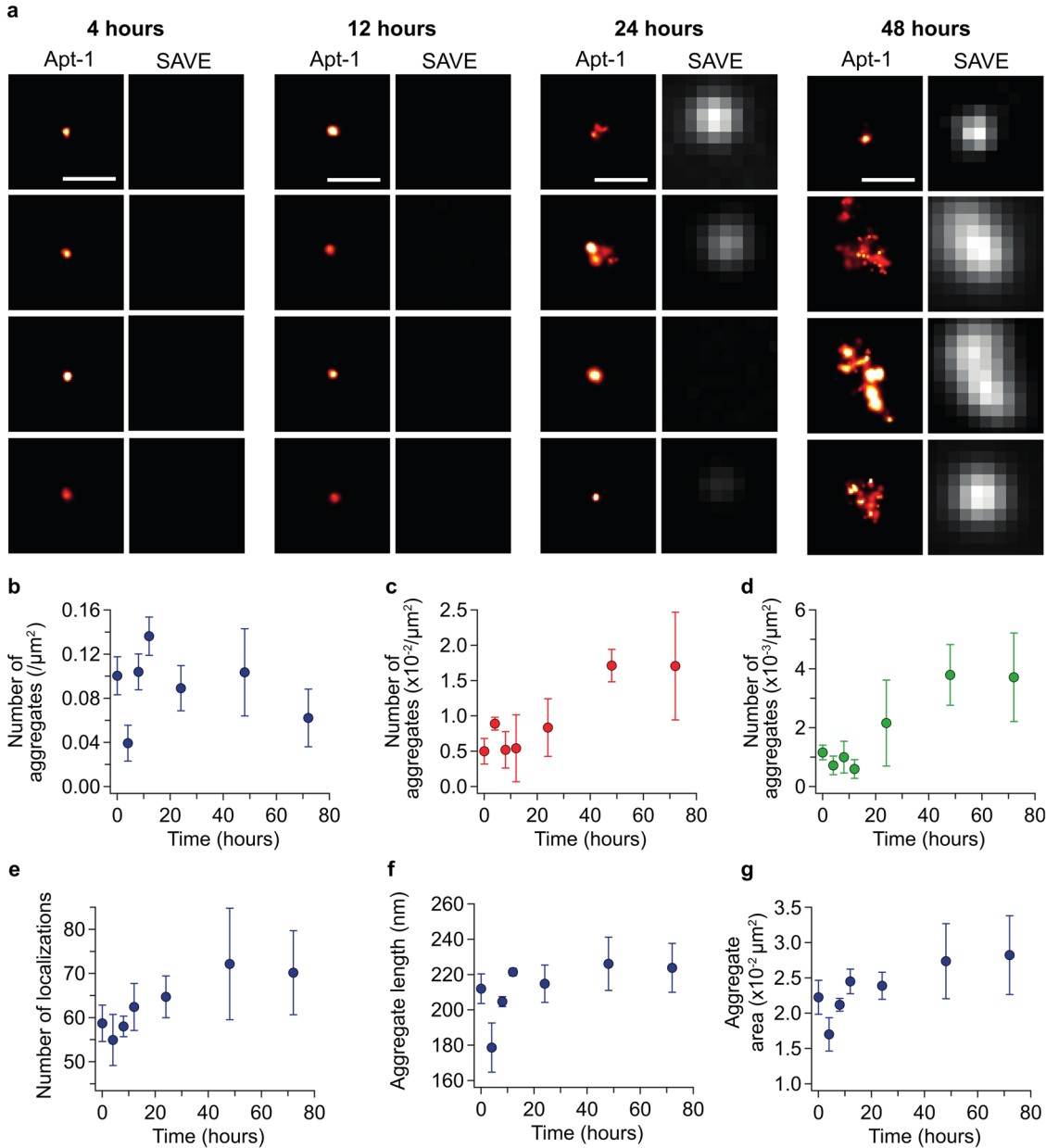

**Fig. 5 Following the aggregation of RRM1-2 using Apt-1 and SAVE imaging. a** Representative Apt-1 SR images (left, red hot) and diffraction-limited SAVE images (right, gray) of RRM1-2 species over a 72 h aggregation assay, scale bar: 500 nm. **b** After an initial decrease in the number of aggregates detected using Apt1 from 0 to 4 h, there is a subsequent increase in their number until 48 h, after which there is a slight decrease as the aggregates get larger in size (**f** and **g**). **c** There is an overall increase in the number of ThT-active aggregates detected using SAVE imaging over 72 h. Approximately 10x fewer aggregates are detected using SAVE imaging compared with SR imaging using Apt1. **d** The number of coincident aggregates, defined as having localizations in the SR image and ThT fluorescence in the SAVE image, increases over time. At 72 h, only 6% of the aggregates detected using Apt1 are ThT-active. **e** The mean number of localizations per cluster increases over 72 h, as does the mean length (**f**) and area (**g**) of each aggregate analyzed using SR microscopy. Data shown are mean ± SD of three independent aggregation reactions. Source data are provided as a Source data file[64].

two-tailed unpaired *t*-test) with 5 times as many aggregates. As a control for binding to off-target amyloidogenic proteins, we investigated to what degree Apt-1 could visualize aggregates composed of Aβ42 and α-synuclein. Neither Apt-1 nor nApt-1 was able to exhibit substantial binding to either protein, meaning that a low number of aggregates could be detected using this approach (Fig. S13). Furthermore, unlike aggregates formed from RRM1-2, there was no significant difference in the number of aggregates detected by Apt-1 and nApt-1, meaning that the few aggregates that could be detected were the consequence of unspecific binding of labeled RNA.

**Employing Apt-1 to track TDP-43 in cells**. To experimentally investigate whether Apt-1 could function as a visualization probe in cells, full-length TDP-43 was expressed fused with eGFP in Hek 293T cells. Concomitantly, we transfected Apt-1 conjugated to the fluorophore Atto590. After 24 h, cells were fixed and visualized by confocal microscopy (Fig. 6). The majority of the cells that internalized the vector for eGFP-TDP-43 over-expression were the same in which Apt-1 was transfected (Figs. 6a and S14; green and red fluorescence appeared to be present in the same cells, hinting at an interaction between the protein and the RNA aptamer). Following TDP-43 distribution, cells were found

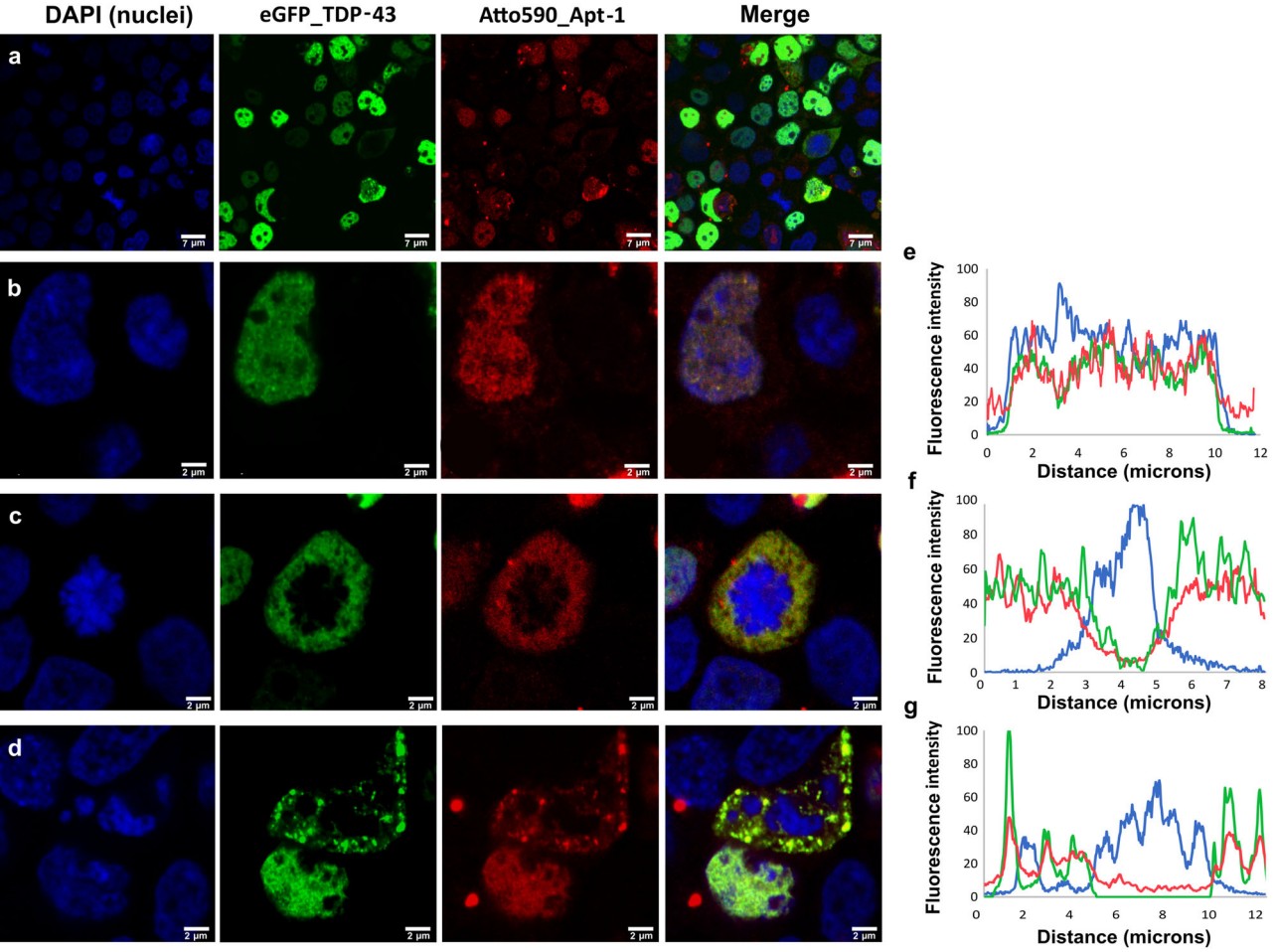

**Fig. 6 Confocal microscopy analysis of TDP-43 and Apt-1 in live mammalian cells.** Blue: DAPI; green: eGFP_TDP-43; red: Apt-1_Atto590. **a** Wide view of Hek293T cells co-transfected with the plasmid for the overexpression of full-length TDP-43 fused to eGFP and the aptamer Apt-1 conjugated to the fluorophore Atto590. **b**–**d** Examples of isolated cells co-transfected as in (**a**), in which the correlation between green and red fluorescence distribution is evident. **e**–**g** Fluorescence profiles of DAPI, eGFP, and Atto590 along a diagonal line drew across the isolated correspondent cells on the left, showing corresponding distribution of the fluorescence signals of eGFP_TDP-43 and Apt-1_Atto590.

in a mixed population composed of elements with nuclear distribution of TDP-43, and cells with mislocalized and/or condensed cytosolic TDP-43 (Fig. 6). This situation recapitulates the pathological TDP-43 behavior[38]. Zooming in on some examples of these cells, the sub-cellular distribution of TDP-43 and Apt-1 was analyzed by imaging (Figs. 6b–d and S15a–f) and fluorescence profiling (Figs. 6e–g and S15g–i) describing pixel by pixel the fluorescence intensity of DAPI, eGFP, and Atto590. Cells with nuclear TDP-43 also displayed nuclear distribution of Apt-1 (Fig. 6b), suggesting an interaction between soluble TDP-43 and the aptamer. The result was confirmed by the fluorescence profile, in which intensities of DAPI, eGFP, and Atto590 correlated (Fig. 6e). Cells with pathological TDP-43 showed either homogenous cytosolic distribution of the protein (Fig. 6c) or the presence of cytosolic TDP-43 aggregates (Figs. 6d and S15b, c). Also for this type of cells, both imaging and fluorescence profiles showed almost perfect overlap between the green fluorescence of TDP-43 and the red fluorescence of Apt-1, proving that Apt-1 can recognize and bind also mislocalized and aggregated TDP-43 (Figs. 6f, g and S15g–i). To quantify the co-localization of the aptamer and its target protein, we measured the Manders' overlap, which determines the co-occurrence of the two fluorescence signals while taking into account pixel intensity in a Z-stack[39]. For the cell shown in Fig. 6c, Manders' overlap was 0.85, while for the cell in Fig. 6d

was 0.89 (these values indicate a co-localization of the fluorescence associated with TDP-43 and Apt-1 of 85% and 89%, respectively). Other examples of Mander's overlap calculations are reported in the SI (Fig. S15). The striking co-occurrence of Apt-1 and TDP-43, resulting into a yellow color derived from the overlap between the red fluorescence of Apt-1 and the green fluorescence of TDP-43, was even more evident in the 3D images derived from the Z-stacks of the analyzed cells (Fig. S16a–f).

We further studied the cellular distribution of nApt-1 in cells overexpressing TDP-43, to validate the specificity of the binding in living cells and confirm the importance of tailored-designed targeting (Fig. 7). For comparison, nApt-1 does not co-localize with TDP-43 in cells, irrespective of sub-cellular localization of the protein (Fig. 7a–d). To quantify this phenomenon, we studied the fluorescence profiles as above described and confirmed a lack of co-distribution of the two fluorophores relative to TDP-43 and nApt-1 (Fig. 7e–g). These results are strengthened by the Manders' overlap values: e.g., 0.25, 0.21, and 0.19 for the three cells reported in Fig. 7. Other examples of the poor or absent co-localization between TDP-43 and nApt-1 are given in Figs. S15d–f and j–l. The corresponding 3D images can be found in Fig. S16g–i.

These analyses confirm that Apt-1 can tightly bind to full-length TDP-43 within the cellular context and suggest that aptamers designed in the reported fashion could be employed as

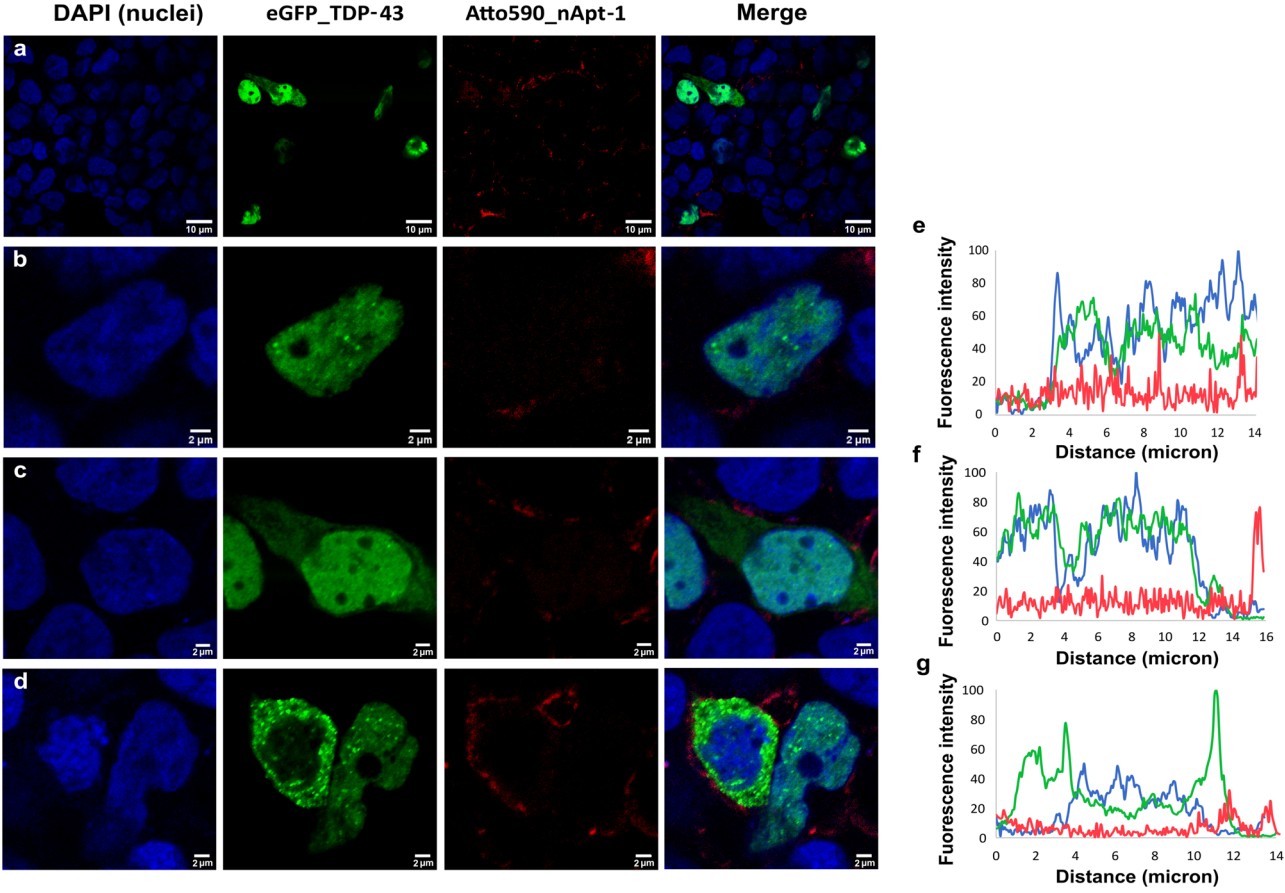

**Fig. 7 Confocal microscopy analysis of TDP-43 and nApt-1 in live mammalian cells.** Blue: DAPI; green: eGFP_TDP-43; red: nApt-1_Atto590. **a** Wide view of Hek293T cells co-transfect with the plasmid for the overexpression of full-length TDP-43 fused to eGFP and the aptamer nApt-1 conjugated to the fluorophore Atto590. **b–d** Examples of isolated cells co-transfected as in (**a**), in which the lack of correlation between green and red fluorescence distribution is evident. **e–g** Fluorescence profiles of DAPI, eGFP, and Atto590 along a diagonal line drew across the isolated correspondent cells on the left, showing how nApt-1_Atto590 mostly localized in the peri-cytoplasmatic regions, irrespective of the distribution of eGFP_TDP-43.

probes for the visualization and identification of their intracellular targets.

## Discussion

In this study, we developed a computational pipeline for de novo design of RNA aptamers in silico. We proved the strength of our approach by generating aptamers against TDP-43, the primary component of cytoplasmic aggregates in ALS patients[38]. Since TDP-43 is a DNA/RNA-binding protein with high tendency to form clinically relevant inclusions, we reasoned that RNA molecules could be exploited to monitor its condensation. Previous work indicates that TDP-43 interactions with RNA molecules span the micromolar–nanomolar affinity range and tight binders contain GU repetitions[16,31].

Aptamer design was carried out through a novel approach based on the *cat*RAPID method to predict protein–RNA interactions[1,2]. Starting from RNA sequences occurring in the human transcriptome and through in silico prioritization, *cat*RAPID identified RNA regions whose GU content increases with the binding propensity for TDP-43, thus mimicking natural TDP-43 binding partners[40]. In our case, iCLIP data were available[41]. Yet, irrespective of data availability, our approach could work equally well in the absence of an initial dataset because the *cat*RAPID algorithm can be used ab initio to generate RNA libraries targeting proteins without available CLIP data[42].

Our computational approach mimics the aptamer selection process of SELEX[1,5,43], a method that functions by iteratively

challenging large populations of nucleic acid sequences to bind a desired target. The ability of *cat*RAPID to predict RNA sequences that bind specifically to a target protein[7,8,44] overcomes the limitations of SELEX, with its need for libraries/reagents, a timeframe of several months and associated costs. *cat*RAPID takes only a few days per target (the typical computational time computed on a standard processor is typically between 2 and 7 days depending on the molecule length). Our approach thus provides a competitive high-throughput method alternative to experimental approaches such as SELEX. In addition, it increases the advantage of the use of aptamers over other biologics, especially in terms of time consumption and costs. Currently, our computational approach does not include a pipeline to predict the effect of specific chemical modifications to enhance RNA stability or avidity towards its target. Steps in these directions have been made[45], but further work is needed. It must also be noted that we focused on a single-domain aptamer and additional computational analysis is required to design appendages for inclusion of functional regions to interact with other parts of TDP-43 or proteins.

We backed up *cat*RAPID with MD simulations that could optimize the recognition and provide information on the leading interactions. This strategy, possibly of limited gain if used ab initio, can instead be particularly helpful when the structure of at least one RNA–protein complex is known and can provide new information that could in turn improve the aptamer design. Other approaches to design aptamers purely based on MD and docking[46,47] cannot be straightforwardly used to perform high-

throughput analyses due to the heavy amount of calculations. Indeed, calculation of $10^6$ mutants on the pool of sequences, as done to estimate the *Protein* and *RNA fitness* scores, is currently impossible by MD or docking. We felt rewarded to see that *cat*RAPID and MD, that are based on different rationales and methodologies, converged in the analysis of Apt-1 and nApt-1. This agreement is, in our opinion, not casual but reflects the correctness of the initial assumptions on the forces that regulate RNA–protein interactions.

We validated our binding propensity predictions demonstrating, by biolayer interferometry, that the aptamers generated by *cat*RAPID bind to the RRM domains of TDP-43 with affinities in the nanomolar to micromolar range. As a first and powerful application, we evaluated the use of our stronger binding aptamer, Apt-1, to visualize and characterize TDP-43 aggregates. In general, the aggregation process can easily be reproduced in vitro, but the determination of the details, such as the morphology of the aggregate and the size at a given time, is highly challenging. This is also true for studies in living cells or tissues, where protein concentrations and kinetics, as well as chemical modifications, differ highly from those reproduced in the test tube. To offer the most complete and realistic view of the protein self-assembly process, both in vitro and in cell, it is necessary to rely on detection strategies that can directly image and characterize also the smaller oligomers of the target protein. Without doubt, the most widely used and simple aggregation detection method in vitro relies on the employment of small dyes such as ThT, specific for the recognition of non-native structures, since they bind significantly better to protein aggregates, rather than small oligomers[48,49]. To demonstrate the efficiency and accuracy of our method compared to ThT staining, we employed the best of our aptamers to visualize RRM1-2 aggregates with SR microscopy as they formed over 72 h. Not only were we able to image individual aggregates, but we could also image them at the nanometer scale, enabling their sizes to be accurately measured. In fact, we achieved an average resolution of 20 nm, and the small size of aptamers ensures that the probes can bind at high densities close to their target epitope. By performing SAVE imaging alongside SR microscopy, we were able to identify two classes of aggregates: a population that was unable to bind ThT; and a second one that was made up of ThT-active aggregates. Interestingly, only 6% of the aggregates detected using Apt-1 were ThT-active, also at later time points, demonstrating the aptamer's ability to identify less mature oligomers that may be important in the disease pathology of TDP-43 and overcoming the limitations of current methods, such as the identification of oligomers of different size and structure. Other most commonly used approaches for the imaging and morphology studies of protein aggregates are transmission electron microscopy (TEM)[50] and atomic force microscopy (AFM)[51]. Both techniques provide qualitative and quantitative information at the nanometer level, but the former is limited in resolution and the latter requires long and complex sample preparation. Our direct labeling of RRM1-2 with Apt-1 allows fast, reproducible, and highly resolved imaging of the earliest species of aggregates formed, as well as all types of larger aggregates. When probed using nApt-1, only few TDP-43 aggregates could be identified by SR microscopy, thus confirming the soundness of our design. We note that the affinities of Apt-1 (nanomolar) and nApt-1 (micromolar) to TDP-43 are not several orders of magnitude apart. This difference, while not surprising for an RBP[52], is highly relevant to study aggregation. Also, Apt-1 binds TDP-43 and not Aβ42 or α-synuclein aggregates, indicating that even if protein condensates attract RNAs[53,54], there is specificity in the interactions involved.

Following the characterization of RRM1-2 in vitro aggregates, we proceeded to image TDP-43 condensates in the cell. Steps towards this direction usually exploit the use of fluorescent proteins, such as GFP, fused to the protein target to identify localization and condensation within the cell. However, these systems often interfere with the condensation process, therefore misrepresenting the results[55]. The use of molecules that recognize specific characteristics of the protein target, such as sequence epitopes or local structures, is a valid alternative. Within this group of molecules, there are antibodies, nanobodies, and aptamers[56]. Compared to the other molecules, aptamers have the great advantage of a smaller molecular weight and, by means of the *cat*RAPID design tool, easier production. Confocal microscopy analysis of cells expressing TDP-43 and transfected with Apt-1 indicate that this aptamer interacts with TDP-43 both in the soluble and condensed forms. By contrast, nApt-1 does not co-localize with the protein, indicating that high affinity towards the target is necessary for successful imaging. We note that Apt-1, designed to interact with RRM1-2, is still sensitive toward oligomers of full-length TDP-43 in the cellular context. This hints at the exposure of its binding epitope also in the aggregated form of the protein. This design strategy gave the advantage of detecting all species of TDP-43, which would have been more difficult with an aptamer designed against the aggregated protein only[57,58]. Thus, exploiting an aptamer that interacts with TDP-43 in the soluble state is effective to monitor the evolution of assemblies over time. For other proteins, it might be important to design aptamers against regions that are available upon aggregation.

In summary, our ability to generate aptamers de novo and use them for microscopy opens a new page in the field of high-resolution molecular imaging to study the evolution of aggregates in tissues. In the future, the method may be used, for instance, to explore pools of samples directly from ALS patients, providing new results at unprecedented resolution. Our aptamers will provide a valuable tool for various detection methods and accelerate our understanding of TDP-43 aggregation in ALS, FTLD, and other neurodegenerative diseases. They might help with the identification of the earliest species of aggregates formed, to understand their spreading and toxicity[59,60], and to design approaches to block them[31]. More in general, the method or its sub-parts (i.e., aptamer design and aptamer-based SR spectroscopy) may also have an important impact on drug development as well as on basic research.

## Methods

**Design of candidate aptamers for TDP-43 using *cat*RAPID**. Our starting point for the *cat*RAPID predictions was a list of more than 3 million RNA fragments identified by iCLIP to bind TDP-43[16]. From these, we selected contiguous sequences enriched over the experimental control and characterized for having interacting nucleotide close to each other (maximum distance of three nucleotides between two peaks a minimum region size of 50 nucleotides). For each interaction site, we computed the iCLIP score defined as the sum of the experimental reads divided by the length of the sequence. We generated a positive set composed of 100 sequences with the highest iCLIP scores. As a negative counterpart, we selected a pool of 100 sequences from the iCLIP data, each having the same length of the positive set, but with contiguous non-interacting nucleotides. To assess *cat*RAPID performances (Fig. 1a), we considered the Area Under the Curve (AUC) of the Receiver Operating Characteristic (ROC) curve generated using iCLIP data and their *cat*RAPID scores. In this analysis, the TDP-43 interaction propensity computed with *cat*RAPID was used to discriminate between interacting (high iCLIP score) and non-interacting (low iCLIP scores) transcriptomic regions. Ranking from moderate- (100 transcriptomic regions with highest iCLIP score compared with 100 transcriptomic regions with the lowest iCLIP score) to strong-signal experimental data (20 transcriptomic regions with the highest iCLIP score compared with 20 transcriptomic regions with the lowest iCLIP score), we found that the AUC increases progressively reaching a maximal value of 0.89, which indicates that *cat*RAPID accurately identifies strong-signal interactions.

Once the overall performances were calculated, the RNA sequences were trimmed to identify the best binders. The top iCLIP score regions (30 sequences) were divided into 10 nucleotide fragments (260 sequences). In our calculations, we selected an aptamer length of 10 nucleotides because the deposited PDB structure of TDP-43 in complex with a UG-rich RNA sequence shows contacts for 10 nucleotides (PDB code 4bs2, GUGUGAAUGAAU)[22]. To prioritize our candidate

RNA aptamers, we calculated the *RNA* and *Protein Fitness* scores. For each aptamer candidate $s$, the *RNA Fitness* score (1), ranging between 0 and 1, was introduced to assess the effect of a random mutation $i$ on the *cat*RAPID interaction propensity[7] for TDP-43, $\pi(s, \text{TDP43})$:

$$RNA\ Fitness = \ell^{-1} \sum_{i=1}^{\ell} \theta[\pi(s, \text{TDP43}) - \pi(i, \text{TDP43})] \qquad (1)$$

Where $\theta[x]$ is the Heaviside step function of $x$: $\theta = 1$ if $x > 0$ and zero otherwise ($\ell = 100$ for all single and double-point mutations of $s$).

Similarly, the *Protein Fitness* score (2) of a sequence $s$, ranging between 0 and 1, evaluates how strong is the *cat*RAPID interaction propensity $\pi$ for TDP-43 in comparison with a protein sequence $i$ having the same length and amino acid composition ($\ell = 100$ proteins are used for each RNA sequence)[9,10].

$$Protein\ Fitness = \ell^{-1} \sum_{i=1}^{\ell} \theta[\pi(s, \text{TDP43}) - \pi(s, i)] \qquad (2)$$

The candidate aptamers were generated considering the ranking of both the *Protein* and *RNA Fitness* score. The selected candidates have positive interaction propensity, *RNA Fitness* score of 1 and *Protein Fitness* > 0.75. The aptamers were named after their *Protein Fitness* score, from 1 (0.99 score) to 6 (0.75 score). We also analyzed how the *Protein* and *RNA Fitness* scores vary upon increasing or decreasing Apt-1 size and found that the highest values are reached at the length of 10 nucleotides (explored range: 6–15 nucleotides), which supports our initial choice of the aptamers' length.

**Structural properties of aptamers interactions investigated by molecular dynamics**. We used SimRNA webserver[27] to generate a set of initial structures for Apt-1 and nApt-1. The aptamers were simulated in water to explore the conformational space and to select extended structures that would preferably interact with TDP-43 based on previous knowledge[22]. Each system was simulated by MD, starting at room temperature and progressively increasing the temperature until the percentage of extended conformers corresponded to at least 10% of the total possible structures. The trajectories were generated using a leapfrog integration algorithm with a time step of integration of 2 fs. The coupling system of the modified Berendsen thermostat was applied with pressure set at 1 bar, following Parrinello-Rahman method[61].

Once the most representative extended conformers were selected through a k-means clusterization of the conformational space, we docked these RNAs with RRM1-2 with HDOCK software, in order to obtain an ensemble of the complex structure[28]. Docking was constrained to the region of RRM1-2 in direct contact with RNA using the information available from the NMR model structure[22].

MD trajectories with length of 2 μs were run in explicit solvent on the docked structures. Simulations were carried out setting the system temperature to 300 K. The equilibrium condition was evaluated based on the root mean square deviation (RMSD). A set of conformers were considered as stable if the RMSD did not vary significantly in the trajectory. Starting from 1 μs, we prolonged the simulation of 1 μs to confirm that the RMSD had small fluctuations (<1 Å) in at least the 60% of the trajectory.

Considering trajectories at equilibrium, we studied a number of properties for Apt-1, nApt-1, and reference NMR oligonucleotide.

We first calculate the number of contacts made by RRM1-2 amino acids of the binding site (BS) with the different RNA structures (Fig. S4). We defined a contact between two residues if at least one atom of the amino acid is at a distance of <4 Å from at least one atom of the nucleotide.

Hydrogen bonds (H-bonds) were calculated using Gromacs default function[62,63]. We calculated the distribution of the H-bond distances between RRM1-2 and the aptamer. In our analysis, we reported a descriptor indicating the probability of finding a H-bond within a distance cutoff of 2.9 Å to measure the most probable distance of the H-bonds for the aptamer-RRM1-2 complex.

As a measure of coordinated motion, we calculated the covariance among the amino acids of RRM1-2-binding site (BS), under the hypothesis that systems with higher binding affinity are characterized by strongly coordinated motions. We built a covariance matrix by measuring how much each amino acid has a motion correlated with the other amino acids withing the BS. The average of all elements in the covariance matrix was then computed.

**Protein purification**. Purification of RRM1, RRM2, and RRM1-2 from kanamycin-resistant pET-Sumo expression vectors with a six-histidine tag (his-tag) and expressed in Rosetta (DE3) *E. coli* cells was carried out as previously described[24]. Briefly, after cell lysis and centrifugation, the protein fractions were resuspended and passed through a nickel-coated resin. Non-His-tagged proteins were washed through and the constructs were eluted with high imidazole concentration (10 mM potassium phosphate buffer pH 7.2, 150 mM KCl, 250 mM imidazole). The eluate was dialyzed and cleaved by Tobacco Etch Virus protease cleavage (1:20 enzyme:protein molar ratio). Following a further affinity purification step with a nickel-coated resin, a HiTrap Heparin column was used to remove nucleic acids. The protein was then eluted with 1.5 M KCl and submitted to size-exclusion chromatography with a HiLoad 16/60 Superdex 75 column. Protein identity and purity were assessed by PAGE. The proteins were flash frozen and stored in low salt buffer (10 mM potassium phosphate pH 7.2, 15 mM KCl) at −80 °C.

**Details of Apt-1 and nApt-1**. Apt-1 and nApt-1 were purchased from Sigma-Merck. The aptamers were conjugated to either biotin or the fluorophore Atto590, according to the type of experiment. For the in-cell studies, selected nucleotides on the RNA sequences were modified with the corresponding fluorinated versions at the 2′ position of the ribose, to increase stability against nucleases. All details relative to the sequences, type, and positions of the aptamers employed in this work can be found in the Supplementary Information (Table S4).

**Quantification of the aptamer binding constants**. Biolayer interferometry experiments were acquired on an Octet Red instrument (ForteBio, Inc., Menlo Park, CA) operating at 25 °C. The binding assays were performed in 10 mM potassium phosphate buffer (pH 7.2) with 150 mM KCl and 0.01% Tween20. Streptavidin-coated biosensors were loaded with 1 μg/ml RNA aptamer modified with biotin on the 3′ end and exposed to increasing protein concentrations varying from 20 nM to 20 μM, according to the strength of the binding. $K_d$ values were estimated by fitting the response intensity (shift in the wavelength upon binding) as a function of the protein concentration, at the steady state. The assay was repeated at least 3 times, each time in triplicate. The reported binding curves are examples of the output of one experiment.

**Protein aggregation assays**. Purified RRM1-2 samples stored at −80 °C were rapidly defrosted and diluted to 20 μM in a high salt buffer (10 mM potassium phosphate buffer pH 7.2, 150 mM KCl). Constructs were subsequently spun at $100,000 \times g$ for 1 h to remove any degraded or aggregated protein. The final protein concentration was assessed by absorbance at 280 nm and adjusted to 15 μM. Protein aggregation was carried out at 37 °C under non-shaking conditions. Aliquots were taken at given time points (0, 4, 8, 12, 24, 48, 72 h) and flash frozen for later analysis. Experiments were repeated three times. Aggregation assays were processed and plotted on the Tecan Magellan Data Analysis software and their statistical analysis was perform with Microsoft Excel 16.6.

**TIRF microscope measurements**. Single-molecule imaging was carried out using a custom-built TIRF microscope, restricting excitation of fluorophores within the sample to 200 nm from the sample-coverslip interface. The fluorophores were excited at either 405 nm (ThT), or 561 nm (Atto590). Collimated laser light at wavelengths of 405 nm (Cobolt MLD 405–250 Diode Laser System, Cobalt, Sweden) and 561 nm (Cobolt DPL561-100 DPSS Laser System, Cobalt, Sweden) were aligned and directed parallel to the optical axis at the edge of a 1.49 NA TIRF Objective (CFI Apochromat TIRF 60XC Oil, Nikon, Japan), mounted on an inverted Nikon TI2 microscope (Nikon, Japan). A perfect-focus system corrected the imaging process for any stage-drift. Fluorescence was collected by the same objective and separated from the TIR beam by a dichroic mirror Di01-R405/488/561/635 (Semrock, Rochester, NY, USA). Collected light was then passed through appropriate filters (405 nm: BLP01-488R-25 (Semrock, NY, USA), 561 nm: LP02-568-RS, FF01-587/35 (Semrock, NY, USA). The emission beam was passed through a 2.5x beam expander and focussed onto an EMCCD camera for image collection (Delta Evolve 512, Photometrics, Tucson, AZ, USA) operating in frame transfer mode (EMGain = 11.5 e⁻/ADU and 250 ADU/photon). Pixel size was 103 nm. Images were recorded with an exposure time of 50 ms with 561 nm illumination (~50 W cm⁻¹), followed by 405 nm excitation (~100 W cm⁻¹). The microscope was automated using the open-source microscopy platform Micromanager (NIH, Bethesda).

**Preparation of imaging probes for PAINT and SAVE imaging**. Atto590-tagged Apt-1 (sequence provided in Supplementary Table S4) was diluted in PBS buffer and used at a final imaging concentration of 1 nM. ThT (Sigma) was dissolved in absolute ethanol (99%) and then diluted in PBS buffer and filtered (0.02 μm filtered, Anotop25, Whatman). The exact concentration was determined by absorbance ($\varepsilon_{421}$ nm = 36,000 M⁻¹ cm⁻¹). The ThT solution was further diluted to 5 μM in the solution of Atto590-tagged Apt-1 for AD-PAINT and SAVE imaging.

**PAINT and SAVE image analysis**. The positions of the transiently immobilized Apt-1 within each frame were determined using the PeakFit plugin (an imageJ/Fiji plugin of the GDSC Single Molecule Light Microscopy package (http://www.sussex.ac.uk/gdsc/intranet/microscopy/imagej/gdsc_plugins) for imageJ using a 'signal strength' threshold of 30 and a precision threshold of 60 nm. The localizations were grouped into clusters using the DBSCAN algorithm in Python 3.8 (sklearn v0.24.2) using epsilon = 1 pixels and a minimum points threshold of 30 to remove random localizations. The clustered localizations were plotted as 2D Gaussian distributions, with a width equal to the precision that they were localized to. To determine the length and areas of each cluster, the localizations were plotted with widths equal to the precision FWHM and were then analyzed using measure module (skimage v0.18.1). The lengths quoted are the maximum measured axis distance. The SAVE images were first thresholded using a value of intensity mean + 2 × S.D., and then analyzed using the measure module (skimage v0.18.1). Aggregate clusters in the AD-PAINT images were to be ThT-active if any of the localizations had ThT signal greater than the threshold value in the corresponding SAVE image. For each sample, three fields of view were imaged.

**Cell preparation for imaging and images analysis**. Human embryonic kidney (HEK) 293T cells were cultured in Dulbecco's modified eagle medium (DMEM) enriched with L-glutamine and kept at 37 °C with 5% $CO_2$. For the microscopy studies, cells were plated on 24-well plates containing coverslips pre-treated with poly-L-lysine. After 24 h, or when confluence was around 65%, cells were co-transfected with 1.5 µg/ml DNA plasmid for TDP-43 overexpression and 1 µg/ml RNA aptamer using the transfection agent Lipofectamine 3000, according to the published protocol (Invitrogen). TDP-43 wild-type gene was cloned downstream to the eGFP gene in a pEGFP C1 mammalian transfection vector; Apt-1 was purchased with the fluorophore Atto590 at its 3′ end and with the cytosines on positions 1 and 9 chemically modified with 2′-Fluoro modification, to increase in-cell stability against nuclease degradation (Supplementary Table S4). nApt-1 was purchased with the fluorophore Atto590 at its 5′ end and with the guanines on positions 1 and 9 chemically modified with 2′-Fluoro modification, to increase in-cell stability against nuclease degradation (Supplementary Table S4). 24 h after transfection, cells were washed with phosphate-buffered saline (PBS) solution and fixed with 4% paraformaldehyde for 10 min at room temperature. After further washes in PBS, cells were permeabilised with 0.1% Triton-X 100 in PBS for 3 min, washed and treated with 0.5 µg/ml 4′,6-diamidino-2-phenylindole (DAPI) solution. After further washing in PBS, coverslips were placed faced down on glass slides using the mounting medium ProLong™ Diamond Antifade Mountant (Invitrogen).

Glass slides with fixed cells were visualized with a Nikon's A1R MP multiphoton confocal microscope, employing the ×60 objective and 3 channel non-descanned detectors. Acquisition and analysis were performed with the Nikon software NIS-Elements Advanced Research version 5.30.02, 64 bit. Fluorescence profiles were defined by drawing a line across specific cells and calculating the fluorescence intensity of the three fluorophores (DAPI, eGFP, and Atto590) pixel by pixel. Z-stacks for selected cells were composed acquiring scans every 0.5 µm for 6 µm above and below the median plane. Through the Z-stacks, the Manders' overlap, which determines the co-occurrence of two selected fluorescence signals while taking into account pixel intensity, was derived by exploring the correlation between green and red fluorescence values. Cells' 3D images were reconstructed from the Z-stacks. Transfection and images acquisitions were repeated 3 times, each time in duplicates. Fluorescence profiling and Mander's overlap were calculated for at least 25 cells per sample.

**Reporting summary**. Further information on research design is available in the Nature Research Reporting Summary linked to this article.

## Data availability

The super-resolution and SAVE image data generated in this study have been deposited in the Zenodo database under accession code 6533779. Not all the calculations for aptamers are publicly available due to the filing of a patent (Italian priority application N. 102022000009500) by E.Z. (IIT), A.A. (IIT and CRG), G.G.T. (IIT, CRG, and ICREA), O.K. (University of Edinburgh), M.H.H. (University of Edinburgh), and A.P. (and King's College London) but they can be granted upon reasonable request under non-disclosure agreement. Source data are provided with this paper.

## Code availability

Predictions for the aptamer sequences were carried out by means of the software "*cat*RAPID omics", version 2.0, available at http://s.tartaglialab.com/page/catrapid_omics2_group (the *RNA Fitness* of Apt-1 is at https://tinyurl.com/y3578hr3 and the *Protein Fitness* of Apt-1 is at https://tinyurl.com/evmybn73). Calculations for sequences <50 nucleic acids are restricted in the webserver due to the filing of a patent (see also Data availability) but they can be provided upon reasonable request under non-disclosure agreement.

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

## Acknowledgements
We thank the "RNA Initiative" at IIT, all members of the M.H.H., A.P., and G.G.T. groups, and especially Fernando Cid Samper. M.H.H. wishes to thank UCB Biopharma and Dr Jim Love for providing funding for the instrument used to generate the super-resolution data in this manuscript. A.P. acknowledges funding from UK DRI (grant REI 3556) and AlzheimerUK (grant ARUK-PG2019B-020). O.K. was supported by a Scottish PhD Research & Innovation Network Traineeships in MND/MS. E.Z. received funding from the Newton fellowship scheme and the MINDED fellowship of the European Union's Horizon 2020 research and innovation program under the Marie Skłodowska-Curie grant agreement No. 754490. K.J. was funded via the BBSRCEastBIO doctoral training program (BB/M010996/1). The research leading to these results was supported by European Research Council [RIBOMYLOME n. 309545 to G.G.T. and ASTRA n. 855923 to G.G.T.], H2020 [IASIS n. 727658 to G.G.T. and INFORE n. 825080 to G.G.T.] and MND [840-791 to G.G.T. and A.P.] projects. The authors would also like to acknowledge the help and support received during confocal images acquisition by the group of Giuseppe Vicidomini at the Molecular Microscopy and Spectroscopy Department of IIT.

## Author contributions
E.Z.: Conceptualization, experimental work in vitro and in cell, data analysis, and writing. O.K.: Conceptualization, experimental work in vitro, data analysis, and writing. E.M.: MD calculations, analysis, and writing. A.A.: *cat*RAPID calculations and data analysis. F.P.P.: MD calculations. J.G.: Data analysis and writing. K.J.: Protein purification and data analysis. D.J.K.: Data analysis and writing. S.C.: Funding acquisition, supervision, and writing. G.R.: Funding acquisition and supervision. S.G.: Funding acquisition and supervision. M.H.H.: Conceptualization, data analysis, funding acquisition, supervision, and writing. A.P.: Conceptualization, data analysis, funding acquisition, supervision, and writing. G.G.T.: Conceptualization, *cat*RAPID calculations, data analysis, funding acquisition, supervision, and writing.

## Competing interests
The authors declare the following competing interests: E.Z. (IIT), A.A. (IIT and CRG), G.G.T. (IIT, CRG, and ICREA), O.K. (University of Edinburgh), M.H.H. (University of Edinburgh), and A.P. (and King's College London) have filed the Italian priority patent application N. 102022000009500 on May 9, 2022, having as subject-matter the technology in the present publication.
