## [Peer Review File · Nature Communications]

REVIEWER COMMENTS

Reviewer #1 (Remarks to the Author):

This manuscript describes the *in silico* RNA aptamer screening against RNA recognition motif of TAR DNA binding protein 43 (TDP-43) using catRAPID algorithm. The authors investigated the structural properties of the interactions of the obtained aptamer, Apt-1 with TDP-43 by Molecular Dynamics and showed reasonable analyses. I think this aptamer screening method is noteworthy. The authors used Apt-1 to visualize the condensation of TDP-43 using super-resolution (SR) microscopy and successfully tracked its condensation process. The experiments are well designed and the obtained results are reasonably interpreted. The manuscript is well organized and the details are sufficiently described. The obtained results and findings are enough significant for many scientists. Therefore, I recommend this manuscript for publication in Nat. Commun., but there are several problems to be considered before publication.

The main problem is, I am not sure whether we can call the obtained oligonucleotides as an aptamer because the TDP-43 is the RNA binding protein. We can regard the Apt-1 as just a best sequence against the RNA binding protein, TDP-43 and the authors should discuss the sequence preference of TDP-43 for binding. The authors should also discuss about the potential and limitation of this methodology as an aptamer screening method.

There are other minor points which should be considered as follows;

1. Line 192, For easy understanding, please describe the sequence of the reference NMR oligonucleotide.

2. I do not think it is easy for many readers of Nat. Commun. to follow the experiments presented in this manuscript since many of those do not know much about SR imaging and DNA-PAINT. I think it is better to add brief explanation of those for easy understanding in the text.

3. In the legend of Figure 5, please explain what each image means.

4. Line 268, why do the authors say the cells in Fig. 5c and 5d are unhealthy. Did they check it by another method? Please add some explanations.

5. In Fig.6 why was such a clear difference of the usefulness of the probes for imaging obtained with Apt-1 with K_d of 0.1 μM and nApt-1 with K_d of 1.5 μM ? The difference of those affinities is just ten times and I do not think we can often get this clear difference in imaging with this not big affinity difference of the probes. It would be much appreciated if you add some discussion on it.

6. In Supplementary Figure S5, for clarity, please describe what left images mean and what right images mean.

7. Supplementary Figure S12, is "NegApt-1" the same as nApt-1?

Reviewer #2 (Remarks to the Author):

This manuscript covers the development of an aptamer (Apt-1) as a molecular tool to investigate TDP-43 condensation characteristics via microscopic studies. They implemented *in silico* and experimental approaches using catRapid, iCLIP along with MD simulations and

BLI to design the aptamer and investigate its binding affinity properties. Whilst the proposed work is significant to enable improved theranostic approaches for ALS, the novelty of the scientific approaches and the depth of scientific discussion to support the findings is significantly low, making this reviewer less enthusiastic about the manuscript in its present form.

Reviewer #3 (Remarks to the Author):

The article "Probing Tdp43 Condensation by an In Silico Designed Aptamer" describes an interdisciplinary research to achieve the definition of novel aptamers to bind TDP-43.

The selection of aptamers is a promising route to target a number of disease-associated proteins. In this work, the authors achieved aptamer sequences by a rational design approach based on in silico methods. This is an outstanding result, considering the affinity obtained with the designed aptamers. Typically this selection is experimentally obtained via SELEX, but the possibility of rationally designing aptamers with specific target properties opens to a large number of new possibilities.

Applications of the designed aptamers have been indeed tested in the work. These include an innovative approach to monitor Tdp-43 condensation using super-resolution microscopy that enabled to identify aggregates of the size of 20-50 nm, a level of resolution that I believe is a breakthrough in the field. In addition, the study also focussed on the condensation of TDP-43 in cell, showing that same designed aptamer recognizes TDP-43 in both soluble and phase-separated states.

I believe this is a ground-breaking research that will have significant impact in the field.

Before publication, however, the authors need to address the following points:

1) It is particularly unique that the same molecule can probe the protein in both the monomeric-soluble and condensed state. I would expect that that most molecules would bind to an aggregate, but very few would bind to the monomer. It would be interesting to test the same aptamer on amyloidogenic proteins (e.d. alpha synuclein or beta amyloid). Indeed properties of the oligomeric aggregates should be common across these systems (i.e. primarily hydrophobicity). Adding such an experiment in vitro would increase the visibility of the work because it would show the uniqueness of this approach.

2) MD simulations need further information. For example it is not assessed if these simulations are converged. Also please add a supplementary table with nr of molecules (including waters), nr of atoms, box sizes etc. More setup information would also be required such as for example electrostatic titration, algorithm for treating bonds (e.g. LINCS, SHAKE etc).

3) in order to characterise better the designed aptamer, and in addition to the MD part, some relatively quick in vitro experiments could be added about the isolated molecule, for example CD spectra.

Reviewer #4 (Remarks to the Author):

In this manuscript, Zacco et al. report two achievements:

(1) Without a single wet-lab experiment they generate aptamers that bind their target protein with nanomolar affinity by a computational pipeline that analyzes pre-existing iCLIP

data.

(2) Without a single bit of optimization and tweaking, an Atto-labeled version of their winning aptamer yields super-resolution images in the 20 nm range.

If proven beyond reasonable doubt, these achievements could be classified as very remarkable and they would justify publication in this journal. However, I am not fully convinced.

Methodologically, the manuscript is quite diverse and interdisciplinary and of potential interest to a varied readership. The description of the methods and the provision of proof and controls is, however, not always according to the standards.

Major issues:

1. RNA aptamers have been around for ~30 years now, and they are typically characterized by complex secondary structures and high specificity. The "aptamers" in the current study are all 10 nt long (smaller than any published aptamer so far), do not appear to have any intrinsic secondary structure and seem to interact with their target in a kind of induced fit / adaptive binding mode. The authors claim very high selectivity, however, when their winning aptamer, Apt-1 is converted in the complementary strand (which is a drastic change to each and every nucleotide), the loss in affinity is only very moderate (factor 15). In typical aptamers, a single point mutation often leads to a loss by a factor of 1000, and double- and triple mutants are often completely dead. This strong binding of Apt-1n is quite unexpected and does not serve as a testimony for specificity.

2. The authors describe their computational ranking and scoring procedures only in a rudimentary way. What did they actually do when they calculate their RNA fitness score? Did they only calculate substitutions (transitions, transversions), or did they also consider deletions, insertions and appendages? The latter is quite important, as one of the major application of aptamers is their use as genetically encoded recognition tags, which requires them appending them to some other RNA. What happens to affinity and specificity when the sequences get longer / shorter?

3. The authors state that they modified Apt-1 with Atto-590, without stating exactly where and how, whether there is a spacer, and how long this is. There is also no information on whether they calculated the influence of this modification on the interaction with the target protein, or did any wet-lab assays. Table S10 (announced on p.31, l. 720 and supposed to give information on the Atto-tagged Apt-1) does not exist. The authors do not perform any in-vitro characterization of this dye-labeled aptamer, and go directly to cells.

4. The binding affinity as such is quite irrelevant for PAINT imaging, what matters is the association and dissociation rate constants. Without having determined those, it is quite a random shot to assume that an arbitrarily chosen aptamer works in PAINT. Given the random nature of this shot, I would certainly like to see appropriate controls, such as the same experiment with labeled Apt-1n, and with a target protein with mutated binding epitope. The authors should also provide an explanation for the very different magnitude (signal height) of the bursts (Fig. 3c)

5. There are also open questions regarding the microscopy images. E.g., Fig. S 10 states "The images show similar distribution of the aptamer and its target protein within living cells". I do not agree. Similarity is far from perfect in rows b and c, and there are a lot of punctae solely in the red channel. What are these?

Minor issues:

6. In Fig. 1 it is not clear what the numbers next to the sequences in panels c and d are supposed to mean.

7. Fig. 3 a and c: I am not sure that it is justified to subtract the RMSF of Apt-1n from that of Apt-1, as these are two entirely different ligands. Additionally, better labeling of the axes is advised, as they are identical but show different curves.

8. Fig. 4a: very confusing. The schematic aptamers shown in this panel are at least 50 nt long and have extensive intrinsic secondary structure, quite the opposite of what the "aptamers" in this study are (10 nt, no secondary structure).

9. Supplementary Table 1 is apparently truncated at the right. 3 columns are missing.

Reviewer #1 (Remarks to the Author):

This manuscript describes the *in silico* RNA aptamer screening against RNA recognition motif of TAR DNA binding protein 43 (TDP-43) using catRAPID algorithm. The authors investigated the structural properties of the interactions of the obtained aptamer, Apt-1 with TDP-43 by Molecular Dynamics and showed reasonable analyses. I think this aptamer screening method is noteworthy. The authors used Apt-1 to visualize the condensation of TDP-43 using super-resolution (SR) microscopy and successfully tracked its condensation process. The experiments are well designed and the obtained results are reasonably interpreted. The manuscript is well organized and the details are sufficiently described. The obtained results and findings are enough significant for many scientists. Therefore, I recommend this manuscript for publication in Nat. Commun., but there are several problems to be considered before publication.

The main problem is, I am not sure whether we can call the obtained oligonucleotides as an aptamer because the TDP-43 is the RNA binding protein. We can regard the Apt-1 as just a best sequence against the RNA binding protein, TDP-43 and the authors should discuss the sequence preference of TDP-43 for binding. The authors should also discuss about the potential and limitation of this methodology as an aptamer screening method.

Here the Reviewer makes a point that we considered very closely. We think that the word *aptamers* is correct for the sequences we generated and tested. TDP-43 is an RNA-binding protein and our RNAs are produced starting from TDP-43 natural targets but are by no means full transcripts. They are *consensus sequences* and thus could be regarded as the starting point for engineering. This is in agreement with the definition of aptamers as “single-stranded oligonucleotides that fold into defined architectures and bind to targets such as proteins”^{1,2}. We believe this would apply to our case.

As requested, we explicitly indicated that TDP-43 interactions with RNAs span the micromolar to nanomolar affinity range and tighter binding is observed for GU-rich transcripts. We showed that our method very well predicts known targets and identifies a strong enrichment of GC-rich RNAs in the pool of prioritized sequences, thus proving that it is not strongly dependent on the original library. As mutagenesis is probed to select aptamers that bind with high affinity and specificity for TDP-43, our procedure represents a SELEX-like procedure *in silico*:

Since TDP-43 is a DNA/RNA-binding protein with high tendency to form clinically relevant inclusions, we reasoned that RNA could be exploited to monitor its condensation. Previous work indicates that TDP-43 interactions with RNA molecules span the micromolar - nanomolar affinity range and tight binders contain GU repetitions^{16,31}.

[...]

*catRAPID identified RNA regions whose GU content increases with the binding propensity for TDP-43, thus mimicking natural TDP-43 binding partners*⁴¹.

The limitations of our method have been added to the Discussion:

Current limitations of our computational approach include the lack of a pipeline to predict the effect of specific chemical modifications to enhance RNA stability. Steps in these directions have been made⁴⁶, but further work is needed. It must also be noted that we focused on a single-domain aptamer and additional computational work would be required to design appendages for inclusion of functional regions to interact with other parts of TDP-43 or proteins.

[...]

This design strategy gave the advantage of detecting all the species of TDP-43, which would have been more difficult with an aptamer designed against the aggregated protein only^{58,98}. Thus, exploiting an aptamer that interacts with the TDP-43 in the soluble state is effective to monitor the evolution of assemblies over time. For other proteins, it might be important to design aptamers to regions that are available upon aggregation.

[...]

approaches to design aptamers purely based on MD and docking^{47,87} cannot be straightforwardly used to perform high-throughput analyses due to the heavy amount of calculations.

There are other minor points which should be considered as follows;

1. Line 192, For easy understanding, please describe the sequence of the reference NMR oligonucleotide.

We included the sequence, as suggested. We are sorry for the omission, we worked on this RNA previously³ and did not realize that we were not giving enough detail in the new context.

In our calculations, we selected an aptamer length of 10 nucleotides because the deposited PDB structure of TDP-43 in complex with a UG-rich RNA sequence shows contacts for 10 nucleotides (PDB code 4bs2, GUGUGAAUGAAU)⁴.

2. I do not think it is easy for many readers of Nat. Commun. to follow the experiments presented in this manuscript since many of those do not know much about SR imaging and DNA-PAINT. I think it is better to add brief explanation of those for easy understanding in the text.

We have now added a brief section on super-resolution microscopy and PAINT on pages 4-5.

The diffraction-limit of light restricts optical microscopy to a resolution of ~250 nm. Recently, numerous techniques, grouped under the umbrella term of super-resolution (SR) microscopy, have been developed to surpass this limit, enabling imaging at a resolution as high as 5 nm. We have previously developed an SR method that makes use of aptamers to image targets with nanometer resolution. Aptamer DNA-PAINT (AD-PAINT), much like DNA-PAINT, is a pointillism-based SR technique employing temporal separation of fluorescent localizations to gain a higher spatial resolution. A short single-stranded “docking” DNA oligonucleotide is attached to a target-specific probe capable of binding to the molecule-of-interest. Visualization of the probe occurs when a second single-stranded “imaging” DNA strand tagged with an organic fluorophore transiently binds to the docking sequence and its location is determined with nanometer precision. The repeated binding and unbinding of multiple probes allows for SR resolution to be generated. AD-PAINT uses RNA or DNA aptamers instead of antibodies as probes. In this study we exploit Apt-1’s transient binding to TDP-43 to mimic the transient binding of the “docking” and “imaging” strands. Thus allowing the use of Apt-1 labelled with an organic fluorophore for SR imaging and eliminating the two-component system of AD-PAINT.

3. In the legend of Figure 5, please explain what each image means.

We have now added more information to the figure legend.

4. Line 268, why do the authors say the cells in Fig. 5c and 5d are unhealthy. Did they check it by another method? Please add some explanations.

We agree with the Reviewer that “unhealthy” is not properly defined in the main text and revised the manuscript following this comment.

Following TDP-43 distribution, cells were found in a mixed population composed of elements with nuclear distribution of TDP-43, and cells with mislocalized and/or condensed cytosolic TDP-43 (Fig. 6). This situation recapitulates the pathological TDP-43 behavior³⁸.

We would like to clarify the reason for our wording. Physiological distribution of TDP-43 is mostly nuclear. Mislocalisation of the protein in the cytosol and aberrant condensation are considered cellular features that recapitulate what observed in ALS patients⁵. Indeed, misbehaviour of TDP-43 is hallmark of this disease. Following the Reviewer’s suggestion, we now simply classify the imaged cells according

to the distribution of TDP-43, which was either nuclear (physiological) or cytosolic (pathological). Cells with cytosolic distribution of the protein frequently showed also condensed TDP-43, as reported for ALS pathology. We have now added a sentence to explain what we mean.

5. In Fig.6 why was such a clear difference of the usefulness of the probes for imaging obtained with Apt-1 with Kd of 0.1 μ M and nApt-1 with Kd of 1.5 μ M? The difference of those affinities is just ten times and I do not think we can often get this clear difference in imaging with this not big affinity difference of the probes. It would be much appreciated if you add some discussion on it.

The Reviewer makes an important point. TDP-43 is a canonical RNA-binding protein (RBP) that preferentially binds to RNAs enriched in GU-content^{3,4}. About 80% of RBPs interact with RNA with affinity that ranges from the micromolar to the nanomolar in near-physiological conditions⁶. Thus, a K_d of 100 nM for positive interactions and 1.5 μ M for negative interactions of TDP-43 is expected. We believe that this difference is not of several orders of magnitude but still relevant for our purposes. We added accordingly an explanation to the main text to clarify this point:

Since TDP-43 is a DNA/RNA-binding protein with high tendency to form clinically relevant inclusions, we reasoned that RNA could be exploited to monitor its condensation. Previous work indicates that TDP-43 interactions with RNA molecules span the micromolar - nanomolar affinity range and tight binders contain GU repetitions^{16,31}.

[...]

When probed using nApt-1, very few TDP-43 aggregates could be identified in SR microscopy, thus confirming the goodness of our design. We note that the affinities of Apt-1 (nanomolar) and nApt-1 (micromolar) to TDP-43 are not several orders of magnitude apart, which is not surprising for an RBP⁵³, but the difference is highly relevant to study aggregation. Also, Apt-1 binds TDP-43 and not A β 42 and α -synuclein aggregates, indicating that even though protein condensates attract RNAs^{54,55}, there is specificity in the interactions involved.

Indeed, when probed using nApt-1, very few aggregates could be identified with AD-PAINT (now included in **Figure 4f,g**; reported below as **Figure 1RL**). By contrast, we observed a significant increase in the number of detectable TDP-43 aggregates when we employed Apt-1. In accordance with these results, our in-cell experiments also show a strong co-occurrence of Apt-1 and TDP-43 (above 80%), as quantified using Mendel's overlap. Contrarily, when the measurement was applied to TDP-43 and nApt-1, the value of Mendel's overlap dropped to ca. 20%.

Figure 1RL (Figure 4 of the manuscript): SR imaging of surface immobilized RRM1-2 aggregates. *a)* Schematic representation of SR imaging. The immobilized RRM1-2 aggregate is transiently bound by an Atto590-tagged Apt1 molecule, the position of which is determined with nanometer precision. This process is repeated to generate a SR image of each aggregate. *b)* Example time montage of an oligomer being imaged using PAINT. Each sub-image is separated by 1s, moving through time from left to right then top to bottom; scale bar: 400 nm. *c)* Intensity profile of the oligomer imaged in *b)*. Individual localizations appear as bursts in intensity that are separated in space and time. *d)* Histogram of precisions of the oligomer imaged in *b)*. Each localization is accurately positioned with a precision of 60 nm or less. *e)* SR image (red hot) and diffraction-limited (gray) images of the aggregate shown in *b)*. Scale bar is 500 nm. *f)* Sample fields of view of clustered RRM1-2 aggregates imaged with Apt-1 and nApt-1. Scale bar 1 μm . *g)* Compared to Apt-1, nApt-1 detects significantly less RRM1-2 aggregates. The data shown are mean \pm SD of 9 fields of view. **** $p < 0.0001$; analyzed by t-test.

6. In Supplementary Figure S5, for clarity, please describe what left images mean and what right images mean.

Thank you for pointing this out. We have now better described the image in the figure legend. This figure is now **Figure S8**.

7. Supplementary Figure S12, is “NegApt-1” the same as nApt-1?

Thank you for detecting this typo. We have rectified and changed NegApt-1 for nApt-1.

Reviewer #2 (Remarks to the Author):

This manuscript covers the development of an aptamer (Apt-1) as a molecular tool to investigate TDP-43 condensation characteristics via microscopic studies. They implemented in silico and experimental approaches using catRapid, iCLIP along with MD simulations and BLI to design the aptamer and investigate its binding affinity properties. Whilst the proposed work is significant to enable improved theranostic approaches for ALS, the novelty of the scientific approaches and the depth of scientific discussion to support the findings is significantly low, making this reviewer less enthusiastic about the manuscript in its present form.

We would like to thank for the comment but it would have been more helpful for us to have a clearer indication of what the Reviewer would like to see given that we have extensively discussed the implications of our work. As for the novelty of the approach, we believe that this is the first time that a purely computationally designed library of aptamers is obtained, tested for their affinity and used for SR microscopy. While individually many of techniques predate this work, this is the first study describing the complete pipeline of a unique new technique for diagnostic and basic science.

Reviewer #3 (Remarks to the Author):

The article "Probing Tdp43 Condensation by an In Silico Designed Aptamer" describes an interdisciplinary research to achieve the definition of novel aptamers to bind TDP-43.

The selection of aptamers is a promising route to target a number of disease-associated proteins. In this work, the authors achieved aptamer sequences by a rational design approach based on in silico methods. This is an outstanding result, considering the affinity obtained with the designed aptamers. Typically this selection is experimentally obtained via SELEX, but the possibility of rationally designing aptamers with specific target properties opens to a large number of new possibilities.

Applications of the designed aptamers have been indeed tested in the work. These include an innovative approach to monitor Tdp-43 condensation using super-resolution microscopy that enabled to identify aggregates of the size of 20-50 nm, a level of resolution that I believe is a breakthrough in the field. In addition, the study also focussed on the condensation of TDP-43 in cell, showing that same designed aptamer recognizes TDP-43 in both soluble and phase-separated states.

I believe this is a ground-breaking research that will have significant impact in the field.

Before publication, however, the authors need to address the following points:

1) It is particularly unique that the same molecule can probe the protein in both the monomeric-soluble and condensed state. I would expect that that most molecules would bind to an aggregate, but very few would bind to the monomer. It would be interesting to test the same aptamer on amyloidogenic proteins (e.d. alpha synuclein or beta amyloid). Indeed properties of the oligomeric aggregates should be common across these systems (i.e. primarily hydrophobicity). Adding such an experiment in vitro would increase the visibility of the work because it would show the uniqueness of this approach.

We thank the Reviewer for highlighting one of the strengths of our approach, that we hope to have sufficiently underlined now with the additional experiments the Reviewer suggested. We employed BLI to determine the binding between monomeric A β 42/ α -synuclein and Apt-1/nApt-1. Indeed, under the same conditions in which we defined the K_d for RRM1-2 of TDP-43 and Apt-1 to be ca. 100 nM and the one of RRM1-2 and nApt-1 to be ca. 1.5 μ M, we detected no binding of Apt-1 (red) nor nApt-1 (yellow) for either of the two amyloidogenic proteins in their monomeric forms. This indicates that the K_d in these cases is at least $>10 \mu$ M (the highest protein concentration tested, chosen to be able to maintain the proteins in their soluble form for the time of the experiment) (**Figure S3** reported below as **Figure 2RL**).

Figure 2RL (Figure S3 of the manuscript). Bi-layer interferometry investigation of the binding between the aptamers and control amyloidogenic proteins. a) Binding curves for Aβ42; b) Binding curves for α-synuclein, red: Apt-1; yellow: nApt-1. At the highest tested concentration of 10 μM no binding was observed for either proteins.

We have also verified potential binding between Apt-1/nApt-1 and the aggregated forms of Aβ42 and α-synuclein, employing SR microscopy. For RRM1-2 aggregates, there was a significantly higher number of aggregates detected using Apt-1 compared with nApt-1. For Aβ42 and α-synuclein, neither Apt-1 nor nApt-1 were able to detect a high number of aggregates. These data have now been included in the SI as **Figure S13 (Figure 6 RL in response to Reviewer 4's comments)**. We added this part to the Results and Discussion:

As a control, we investigated the interaction of Apt-1 and of nApt-1 with two more amyloidogenic proteins in their soluble forms: Aβ42 and α-synuclein (Fig. S3). Neither protein shows any binding ability for Apt-1 nor nApt-1 within the tested conditions, emphasizing the specificity of Apt-1 towards TDP-43.

[...]

To compare the sequence specificity of Apt-1 to TDP-43, sample images of RRM1-2 aggregates were taken with Apt-1 and nApt-1. The number of aggregates detected with Apt-1 was significantly higher than with nApt-1 (Fig. S13; p -value < 0.0001 , two-tailed unpaired t -test) with 5 times as many aggregates. As a control for binding to off-target amyloidogenic proteins, we investigated to what degree Apt-1 could visualize aggregates composed of Aβ42 and α-synuclein. Neither Apt-1 nor nApt-1 were able to exhibit substantial binding to either protein, meaning that a very low number of aggregates could be detected using this approach (Fig. S13). Furthermore, unlike aggregates formed from RRM1-2, there was no significant difference in the number of aggregates detected by Apt-1 and nApt-1, meaning that the few aggregates that could be detected were due to non-specific binding of labeled RNA.

2) MD simulations need further information. For example it is not assessed if these simulations are converged. Also please add a supplementary table with nr of molecules (including waters), nr of atoms, box sizes etc. More setup information would also be required such as for example electrostatic titration, algorithm for treating bonds (e.g. LINCS, SHAKE etc).

The Reviewer is fair. We have now abundantly implemented the description of our MD simulations and hope to have fully answered the Reviewer's request. A table with the details of the analyses have been added to the SI (**Table S2**) and is also reported below:

	Apt-1 and nApt-1 systems
Simulation time step	2 femtoseconds
Acquisition time step	100 picoseconds
algorithm of integration	Leap-frog
MD software	GROMACS
Force field	AMBER99
Total time	1 microsecond*
Reference Temperature	300K
Temperature coupling	modified Berendsen
Reference Pressure	1 bar
Pressure coupling	Parinello-Rahman
box size	1 nm distance from structure
constraint - algorithm	LINCS
RMSD fitting reference	Protein + RNA
RMSD calculation reference	Protein + RNA

* The time is increased by steps of 1 microsecond in the case the MD simulation is not converged in at least the 60% fraction of the last microsecond.

	Apt-1	nApt-1
Total number of atoms	104808	72955
Number of water molecules	33902	23282

To reply to the Reviewer's observation on the convergence of the two simulated systems, we include here a better description of the method and report it also in the paper. We performed a Root Mean Square Deviation (RMSD) analysis on all atoms of the protein-RNA complex without ions and solvent, after fitting the structure on the same atoms. As shown in the figure below (**Figure S4**), the two systems are characterized by a final phase of equilibrium. In accordance with the strategy described in the manuscript, we incremented the dynamics by 1 microsecond at the time, until at least 60% of MD simulations converged. Since the system with nApt-1 is characterized by an initial fluctuating RMSD

value, its simulation needed to be extended to 3 microseconds, while the RMSD of the system in the presence of Apt-1 reaches stability within the first 1 microsecond of simulation (see **Figure 3RL**).

Figure 3RL (Figure S4 of the manuscript). MD trajectories were run in explicit solvent. Simulations were carried out setting the system temperature to 300 K. The equilibrium conditions were evaluated based on the Root Mean Square Deviation (RMSD). Starting from 1 μ s, we prolonged the simulation of 1 μ s to confirm that the RMSD had small fluctuations (<1 Å) in at least the 60% of the trajectory.

3) in order to characterise better the designed aptamer, and in addition to the MD part, some relatively quick in vitro experiments could be added about the isolated molecule, for example CD spectra.

We further characterized Apt-1 using circular dichroism and confirmed its linear nature, in agreement with structural predictions. (**Figure S5** and below). The binding between TDP-43 and short linear RNA is also reported in the NMR structure described in 4BS2 (see **Figure 4RL**).

Figure 4RL. Structural analysis of Apt-1. a) Circular dichroism analysis of the aptamer Apt-1, in which the spectrum displays the typical maxima and minima of an RNA sequence without any noteworthy secondary structure; b) Prediction of the structure of Apt-1 by means of the algorithm “RNAfold”, showing very poor base-pair probabilities and emphasizing the linear nature of the sequence.

Reviewer #4 (Remarks to the Author):

In this manuscript, Zacco et al. report two achievements:

- (1) Without a single wet-lab experiment they generate aptamers that bind their target protein with nanomolar affinity by a computational pipeline that analyzes pre-existing iCLIP data.
- (2) Without a single bit of optimization and tweaking, an Atto-labeled version of their winning aptamer yields super-resolution images in the 20 nm range.

If proven beyond reasonable doubt, these achievements could be classified as very remarkable and they would justify publication in this journal. However, I am not fully convinced.

Methodologically, the manuscript is quite diverse and interdisciplinary and of potential interest to a varied readership. The description of the methods and the provision of proof and controls is, however, not always according to the standards.

Major issues:

1. RNA aptamers have been around for ~30 years now, and they are typically characterized by complex secondary structures and high specificity. The “aptamers” in the current study are all 10 nt long (smaller than any published aptamer so far), do not appear to have any intrinsic secondary structure and seem to interact with their target in a kind of induced fit / adaptive binding mode. The authors claim very high selectivity, however, when their winning aptamer, Apt-1 is converted in the complementary strand (which is a drastic change to each and every nucleotide), the loss in affinity is only very moderate (factor 15). In typical aptamers, a single point mutation often leads to a loss by a factor of 1000, and double- and triple mutants are often completely dead. This strong binding of Apt-1n is quite unexpected and does not serve as a testimony for specificity.

Here, the Reviewer indicates something extremely relevant, which requires proper explanation in the main text to clarify issues concerning what type of results we were expecting from our molecules. TDP-43 is a canonical RBP that binds to single-stranded RNAs enriched in GU-content^{3,4}. For high-complexity sequences, a K_d of 100 nM for positive interactions and 1.5 μ M for negative interactions is in the spectrum of expected affinities. We are aware that sequences such as polyalanine would display lower interaction affinities towards TDP-43⁷. Yet low-complexity sequences, including single, double and triple nucleic acid repetitions, are known to target several RBPs⁸, and could introduce additional effects in the cellular experiment, such as aggregation⁹. Thus, the reverse complementary sequence represents a good control, since has the same sequence complexity as Apt-1. As a matter of fact, the ten-fold difference in K_d between Apt-1 and nApt-1 is relevant for our purposes. The in-cell experiments show a co-occurrence of Apt-1 and TDP-43 (above 80%), as quantified using Mendel’s overlap. By contrast, when the measurement is applied to TDP-43 and nApt-1, the value of Mendel’s overlap drops to ca. 20%.

Motivated by the Reviewer's comment, we performed additional analyses. As shown in new experiments (see **Figure 1RL** in response to **Reviewer 1**), when nApt-1 is used for probing, very few aggregates could be identified (new **Figure 4f,g**). By contrast, we observed a significant increase in the number of detectable TDP-43 aggregates when Apt-1 was employed. Thus, the observed difference in K_d between Apt-1 and nApt-1 indicates that the two molecules behave differently in vitro. This has been added to the Discussion:

We note that the affinities of Apt-1 (nanomolar) and nApt-1 (micromolar) to TDP-43 are not several orders of magnitude apart, which is not surprising for an RBP⁵³, but the difference is highly relevant to study aggregation. Also, Apt-1 binds TDP-43 and not A β 42 and α -synuclein aggregates, indicating that even though protein condensates attract RNAs^{54,55}, there is specificity in the interactions involved.

Finally, we would like to mention that the short size of our aptamers allows further developments to attach probes for visualization or additional molecules for delivery (see answer to **point n.2** below).

2. The authors describe their computational ranking and scoring procedures only in a rudimentary way. What did they actually do when they calculate their RNA fitness score? Did they only calculate substitutions (transitions, transversions), or did they also consider deletions, insertions and appendages? The latter is quite important, as one of the major application of aptamers is their use as genetically encoded recognition tags, which requires them appending them to some other RNA. What happens to affinity and specificity when the sequences get longer / shorter?

02/28/2022 14:25:00 Accordingly, we provided more accurate descriptions of the *RNA Fitness* and *Protein Fitness* scores:

To prioritize our candidate RNA aptamers, we calculated the RNA Fitness and Protein Fitness scores. For each aptamer candidate s , the RNA Fitness score, ranging between 0 and 1, was introduced to assess the effect of a random mutation i on the catRAPID interaction propensity⁷ for TDP-43, $\pi(s, TDP43)$:

$$RNA\ Fitness = \ell^{-1} \sum_{i=1}^{\ell} \theta[\pi(s, TDP43) - \pi(i, TDP43)]$$

Where $\theta[x]$ is the Heaviside step function of x : $\theta = 1$ if $x > 0$ and zero otherwise ($\ell = 100$ for all single and double-point mutations of s).

Similarly, the Protein Fitness score of a sequence s , ranging between 0 and 1, evaluates how strong is the catRAPID interaction propensity π for TDP-43 in comparison with a protein sequence i having the same length and amino acid composition ($\ell = 100$ proteins are used for each RNA sequence)^{9,10}.

$$\text{Protein Fitness} = \ell^{-1} \sum_{i=1}^{\ell} \theta[\pi(s, \text{TDP43}) - \pi(s, i)]$$

The candidate aptamers were generated considering the ranking of both the Protein Fitness and RNA Fitness score. The selected candidates have positive interaction propensity, RNA Fitness score of 1 and Protein Fitness > 0.75. The aptamers were named after their Protein Fitness score, from 1 (0.99 score) to 6 (0.75 score).

Regarding the size of the aptamer, we exploited the information of the 4bs2 PDB structure that an RNA molecules in complex with TDP-43 shows contacts for 10 nucleotides⁴. In agreement with other reports^{3,4}, the PDB structure also indicates that the RNA bound to TDP-43 is single-stranded. Since >90% of 10 nucleotides sequences are prone to be single-stranded¹⁰, we reasoned that among them there could be good TDP-43 aptamers. Accordingly, we added to the main text:

The length of the window was set according to the number of contacts that an RNA oligonucleotide establishes with the RRM domains of TDP-43 in a NMR structure (PDB 4bs2; **Fig. 1b; Online Methods**)²². Importantly, a size of 10 nucleotides ensures that most of the fragments are single-stranded, which is a requirement for TDP-43 binding^{23,25}.

As suggested by the Reviewer, we also considered shorter sequences contained within Apt-1 and extensions of it, in both the natural and artificial context.

Sequence	Raw Score	RNA Fitness	Protein Fitness
CGGUGU	0.37	0.58	0.28
GUUGCU	-1.05	0.39	0.20
CGGUGUU	1.10	0.79	0.47
UGUUGCU	-0.31	0.57	0.37
CGGUGUUG	1.96	0.86	0.95
GUGUUGCU	-0.22	0.55	0.36
CGGUGUUGC	1.76	0.80	0.46
GGUGUUGCU	0.00	0.55	0.38
CGGUGUUGCU	1.96	1.00	0.99
CGGUGUUGCUU	1.03	0.87	0.50
UCGGUGUUGCU	0.45	0.79	0.50
AUCGGUGUUGCU	-1.15	0.53	0.36
CGGUGUUGCUUG	1.16	0.85	0.46
CGGUGUUGCUUGC	1.92	0.92	0.88
GAUCGGUGUUGCU	0.81	0.77	0.44
GGAUCGGUGUUGCU	0.10	0.76	0.37
CGGAUCGGUGUUGCU	0.50	0.65	0.31

In the first analysis, we found that Apt-1 shows the highest RNA Fitness and Protein Fitness. We identified CGGUGUUG and CGGUGUUGCUUGC as highly ranking, but their RNA Fitness and Protein Fitness scores are inferior to those of Apt-1. We are very excited about this result and feel most thankful to the Reviewer for suggesting it. Accordingly, we added to the text:

We also analyzed how the Protein Fitness and RNA Fitness scores vary upon increasing or decreasing Apt-1 size and found that the highest values are reached at the length of 10 nucleotides (explored range: 6-15 nucleotides), which supports our initial choice of the aptamers' length.

In the second analysis, we considered extensions of Apt-1 in the artificial context, meaning that nucleotides were progressively added to Apt-1 (**Figure 5RL**). Given the exponential growth of sequences, we restricted the size of fragments to the range 11-15 nucleotides. As the length increases, the amount of structured (double-stranded) RNAs that do not bind to TDP-43 grows. Aptamers with RNA Fitness higher than 0.99 and Protein Fitness higher than 0.95 (i.e., in the range of Apt-1) decreased progressively at each iteration. Yet, considering that 16 out of 418 single-stranded RNAs meet these criteria for aptamers of 15 nucleotides, we conclude that longer molecules can be indeed identified to bind TDP-43. Thus, we think that appendages could be added, but the structural context should be controlled to avoid that the interaction with TDP-43 is altered. Further work is needed depending on the experimental application of interest.

Figure 5RL. By progressively adding nucleotides to Apt-1, we monitored the distribution of 'strong candidates' (RNA fitness higher than 0.99 and Protein Fitness higher than 0.95). Despite the increase of double-stranded regions with longer RNAs, strong candidates could be identified (16 single-stranded RNAs).

3. The authors state that they modified Apt-1 with Atto-590, without stating exactly where and how, whether there is a spacer, and how long this is. There is also no information on whether they calculated the influence of this modification on the interaction with the target protein, or did any wet-lab assays. Table S10 (announced on p.31, l. 720 and supposed to give information on the Atto-tagged Apt-1) does not exist. The authors do not perform any in-vitro characterization of this dye-labeled aptamer, and go directly to cells.

Following up on the Reviewer comment, we have added a table indicating position and nature of the modifications for each of the RNA sequences employed in this study. We did not experimentally determine the effect of the Atto590 modification on Apt-1's binding affinity towards RRM1-2. Yet, the position of the fluorophore on the RNA (3') is the same used for conjugating the biotin, necessary for the BLI experiments. Whilst we cannot rule out any minor effect, replacing biotin with Atto590 did not prevent it from binding to and being used to image RRM1-2 in SR microscopy experiments, or TDP-43 in cells. Conversely, Atto590-modified nApt-1 could not be used to detect RRM1-2 nor TDP-43.

4. The binding affinity as such is quite irrelevant for PAINT imaging, what matters is the association and dissociation rate constants. Without having determined those, it is quite a random shot to assume that an arbitrarily chosen aptamer works in PAINT. Given the random nature of this shot, I would certainly like to see appropriate controls, such as the same experiment with labeled Apt-1n, and with a target protein with mutated binding epitope. The authors should also provide an explanation for the very different magnitude (signal height) of the bursts (Fig. 3c)

The Reviewer is correct in saying that the association and dissociation rate constants are important for PAINT-based imaging methods. However, there is a large range of rates that allow for suitable exchange. If the affinity is high (i.e. high association and low dissociation rate constant), then the exchange rate is low and a longer data acquisition time is required; for example, for super-resolution imaging using the picomolar affinity amyloid binding dye PFTAA, 35,000 frames were required to image fibrils¹¹. Conversely, if the affinity is low (low association, high dissociation rates), the emitter will reside for a shorter period of time, and exchange is rapid; for example, to image fibrils using the dye Nile Red, we imaged for <10,000 frames¹². Indeed, one of the advantages of PAINT is that the imaging conditions can be modified to facilitate a wide range of binding kinetics (the exposure time of the camera, the concentration of the fluorescent probe, the length of time the sample is imaged for).

We have now compared the imaging of fibrils of RRM1-2 with Apt-1 and nApt-1 and have included the results in **Figure 4** (see **Figure 1RL**). We have shown the specificity of Apt-1 to RRM1-2 by imaging alpha-synuclein and amyloid-beta aggregates, and we observe a low number of aggregates (**Figure S13**), reported below as **Figure 6RL**). This has been added in the Results section:

Figure 6RL (Figure S13): Aggregate detection of TDP-43, α -synuclein and A β 42 with Apt-1 and nApt-1 . a) Sample fields of view of clustered aggregates imaged with Apt-1 for respective proteins. Scale bar is 5 μm . b) Plot of mean cluster number per μm^2 showing a significant difference in the detection of aggregates with Apt-1 and nApt-1 only for RRM1-2. The data shown are means \pm SD of 9 fields of view. ** $p < 0.0001$, ns $p > 0.05$; analyzed by *t*-test.**

Due to the stochastic nature of fluorescence, there is always a variation in the intensity of each burst (for example, due to fluorophores absorbing/emitting different number of photons, the fluorophore's environment, the fluorophores position in the evanescent field etc.). This can be observed in Figure 1 of the original publication describing PAINT¹³, and also in Figure 1 of our original publication on Aptamer-DNA-PAINT¹⁴. This is also true in other single-molecule localization microscopy approaches and is evident in the broad distributions of localization precisions.

5. There are also open questions regarding the microscopy images. E.g., Fig. S 10 states “The images show similar distribution of the aptamer and its target protein within living cells”. I do not agree. Similarity is far from perfect in rows b and c, and there are a lot of punctae solely in the red channel. What are these?

We thank the Reviewer for this point, which allowed us to clarify the relevance of our results. In our analysis, we firstly examined the distribution of TDP-43_eGFP (green) in the cell population, and we then determined whether, within the same pixels, we could also identify the fluorescence of the aptamers tagged with Atto590 (red). We quantified the degree of colocalization of TDP-43 and Apt-1 using Mendel's overlap and found it to be above 80% for all tested cells. This result indicates that ca. 80% of all pixels showing green fluorescence also displayed red fluorescence. When the colocalization

measurement was applied to nApt-1, this value dropped to ca. 20%. The fluorescence distribution studies confirmed the same results (see **Figure 7RL**).

In some occasions it is possible to observe red deposits where the green of TDP-43 is not present. We believe that these punctuates are partially degradation products (e.g. exosomal RNA degradation within nucleoli) but mostly they represent a limitation of the technique. The Reviewer may refer to the image below, in which we show the selection process we put in place to choose, among the commercially available transfection reagents, the one that gave us the most satisfying results:

Figure 7RL. Screen of transfection reagents. In our screening, we assessed which reagent could be the most efficient considering transfection and reduction of deposits. Lipofectamine3000 was found to be the best choice for our in-cell studies.

This investigation revealed that Lipofectamine3000 is the transfection reagent that gave the highest transfection efficiency and promotes the partial formation of the red deposits, albeit less than other

reagents. The red deposits we sometimes observe in the cells are consistent with what detected during the process of choosing the transfection reagent. We do not believe they influence the results in any way.

Minor issues:

6. In Fig. 1 it is not clear what the numbers next to the sequences in panels c and d are supposed to mean.

We have now clarified the legend.

7. Fig. 3 a and c: I am not sure that it is justified to subtract the RMSF of Apt-1n from that of Apt-1, as these are two entirely different ligands. Additionally, better labeling of the axes is advised, as they are identical but show different curves.

We thank the Reviewer for this comment. While disassembling the plot in two separate ones, we realized that the interpretation of our results could be made more straightforward thanks to the Reviewer's suggestion. Therefore, we updated **Figure 3** of the manuscript. We observed that both Apt-1 and nApt-1 form contacts with amino acids 180 and 224, which are bound to the RNA in the reference NMR model structure (PDB 4BS2, marked with green boxes at the top of the image; **Figure 8RL**). Apt-1 more frequently contacts amino acids 104-112,135-139,144-150, 194-199, 255-261 and 263-264 (red boxes and regions highlighted in red in the images on the right), also reported in the NMR model, while nApt-1 interacts with amino acids 140-143,165,186-191, 262-263 and 267 (blue boxes and regions highlighted in blue in the images on the right) that are not present in the NMR model. The RMSF shows that nApt-1 binds less well, as the contacts are formed in mobile regions (amino acids 144 and 267, corresponding to loops), whereas Apt-1 interacts with elements with lower flexibility (amino acids 135, 145, 255 and 263). Accordingly, we added to the main text:

*Both aptamers form contacts with amino acids 180 and 224 that are also present in the reference NMR model structure (PDB 4bs2; **Fig. 3a**). Yet, Apt-1 interacts more frequently with amino acids 104-112,135-139,144-150, 194-199, 255-261 and 263-264 (**Fig. 3a**), as reported in the NMR model, while nApt-1 interacts with amino acids 140-143,165,186-191, 262-263 and 267 (**Fig. 3a**) that are not involved in the binding in the NMR model. The Root Mean Square Fluctuation (RMSF; **Fig. 3b; Online Methods**) shows that nApt-1 binds less well because the contacts are formed in highly mobile regions (amino acids 144 and 267, corresponding to loops), whereas Apt-1 interacts with elements with lower flexibility (amino acids 135, 145, 255 and 263, **Fig.3b-d**).*

Figure 8RL (Figure 3 of the manuscript). MD characterization of Apt-1 and nApt-1 interactions with RRM1-2. a) Contact Frequency computed along Apt-1 and nApt-1 MD trajectories; The contacts observed in the NMR model structure are marked with green boxes at the top of the image. Red is used to indicate contacts more frequent for Apt-1 and blue contacts that are more frequent for nApt-1; b) Root Mean Square Fluctuation computed along Apt-1 and nApt-1 MD trajectories; The colour of the boxes follows the same as in panel a; c) Structural representation of residues with stable contacts in complex with Apt-1 (RRM1-2 representative configuration at equilibrium shown; colours correspond to panels a and b); d) Structural representation of residues with stable contacts in complex with Apt-1 (RRM1-2 representative configuration at equilibrium shown; colours correspond to panels a and b).

8. Fig. 4a: very confusing. The schematic aptamers shown in this panel are at least 50 nt long and have extensive intrinsic secondary structure, quite the opposite of what the “aptamers” in this study are (10 nt, no secondary structure).

The Reviewer is right. This is because we have used a more conventional drawing of the AD-PAINT technique. We have now replaced this image with one that better represent our specific system.

9. Supplementary Table 1 is apparently truncated at the right. 3 columns are missing.

We apologize, we did not realize this during PDF conversion. We have now rectified as pointed out.

References

1. Keefe, A. D., Pai, S. & Ellington, A. Aptamers as therapeutics. *Nat Rev Drug Discov* **9**, 537–550 (2010).
2. Zhang, Y., Lai, B. S. & Juhas, M. Recent Advances in Aptamer Discovery and Applications. *Molecules* **24**, 941 (2019).
3. Zacco, E. *et al.* RNA as a key factor in driving or preventing self-assembly of the TAR DNA-binding protein 43. *J. Mol. Biol.* **431**, 1671–1688 (2019).
4. Lukavsky, P. J. *et al.* Molecular basis of UG-rich RNA recognition by the human splicing factor TDP-43. *Nat Struct Mol Biol* **20**, 1443–1449 (2013).
5. Suk, T. R. & Rousseaux, M. W. C. The role of TDP-43 mislocalization in amyotrophic lateral sclerosis. *Molecular Neurodegeneration* **15**, 45 (2020).
6. Yang, X., Li, H., Huang, Y. & Liu, S. The dataset for protein–RNA binding affinity. *Protein Sci* **22**, 1808–1811 (2013).
7. Kuo, P.-H., Doudeva, L. G., Wang, Y.-T., Shen, C.-K. J. & Yuan, H. S. Structural insights into TDP-43 in nucleic-acid binding and domain interactions. *Nucleic Acids Res* **37**, 1799–1808 (2009).
8. Jolma, A. *et al.* Binding specificities of human RNA-binding proteins toward structured and linear RNA sequences. *Genome Res* **30**, 962–973 (2020).
9. Cid-Samper, F. *et al.* An Integrative Study of Protein-RNA Condensates Identifies Scaffolding RNAs and Reveals Players in Fragile X-Associated Tremor/Ataxia Syndrome. *Cell Rep* **25**, 3422–3434.e7 (2018).
10. Lorenz, R. *et al.* ViennaRNA Package 2.0. *Algorithms for Molecular Biology* **6**, 26 (2011).
11. Ries, J. *et al.* Superresolution Imaging of Amyloid Fibrils with Binding-Activated Probes. *ACS Chem. Neurosci.* **4**, 1057–1061 (2013).
12. Bongiovanni, M. N. *et al.* Multi-dimensional super-resolution imaging enables surface hydrophobicity mapping. *Nat Commun* **7**, 13544 (2016).
13. Sharonov, A. & Hochstrasser, R. M. Wide-field subdiffraction imaging by accumulated binding of diffusing probes. *PNAS* **103**, 18911–18916 (2006).
14. Whiten, D. R. *et al.* Nanoscopic Characterisation of Individual Endogenous Protein Aggregates in Human Neuronal Cells. *ChemBioChem* **19**, 2033–2038 (2018).

REVIEWERS' COMMENTS

Reviewer #1 (Remarks to the Author):

I think the authors properly revised their manuscript according to all the reviewers' comments and it is now ready for publication in Nature Communications.

Reviewer #3 (Remarks to the Author):

The revised form of the article "Probing Tdp43 Condensation using an In Silico Designed Aptamer" has addressed all my previous points.

The authors should be congratulated for their effort into improving the quality and clarity of their manuscript.

Reviewer #1 (Remarks to the Author):

I think the authors properly revised their manuscript according to all the reviewers' comments and it is now ready for publication in Nature Communications.

We are very grateful for the comments. They allowed us to improve the clarity of the work.

Reviewer #3 (Remarks to the Author):

The revised form of the article "Probing Tdp43 Condensation using an In Silico Designed Aptamer" has addressed all my previous points.

The authors should be congratulated for their effort into improving the quality and clarity of their manuscript.

Thank you very much. The remarks of the Reviewer really helped us to make the manuscript better!